

# Contribution of understorey vegetation and soil processes to boreal forest isoprenoid exchange

Mari Mäki[1], Jussi Heinosalo[2], Heidi Hellén[3], Jaana Bäck[1]

5    [1]Department of Forest Sciences, P.O. Box 27, FI-00014 University of Helsinki, Finland.

[2]Department of Food and Environmental Sciences, P.O. Box 66, FI-00014 University of Helsinki, Finland.

[3]Finnish Meteorological Institute, P.O. Box 503, FI-00101 Helsinki, Finland.

10    Keywords: BVOCs, boreal soil, microbial activity, mycorrhizal fungi, vegetation, trenching

*Correspondence to*: Mari Mäki (mari.maki@helsinki.fi)



**Abstract.** Boreal forest floor emits biogenic volatile organic compounds (BVOCs) from the understorey vegetation and the heterogeneous soil matrix, where the interactions of soil organisms and soil chemistry are complex. Earlier studies have focused on determining the net exchange of VOCs from the forest floor. Our study takes one step forward, with the aim of separately determining whether

the photosynthesized carbon allocation to soil affects the isoprenoid production by different soil organisms, i.e. decomposers, mycorrhizal fungi, and roots. In each treatment, photosynthesized carbon allocation through roots for decomposers and mycorrhizal fungi was controlled by either preventing root ingrowth (50μm mesh size) or the ingrowth of roots and fungi (1μm mesh) into the soil volume, which is called the trenching approach. We measured isoprenoid fluxes using dynamic (steady-state

flow-through) chambers from the different treatments. We also aimed to analyze how important the understorey vegetation is as a VOC sink. Finally, we constructed a statistical model based on prevailing temperature, seasonality, trenching treatments, understory vegetation cover, above canopy photosynthetically active radiation (PAR), soil water content, and soil temperature to estimate isoprenoid fluxes. The final model included parameters with a statistically significant effect on the

isoprenoid fluxes. Our results show that the boreal forest floor emits monoterpenes, sesquiterpenes, and isoprene. Monoterpenes were the most common group of emitted isoprenoids, and the average flux from the non-trenched forest floor was 23 $\mu g\ m^{-2}\ h^{-1}$. Our results show that different biological factors, including litterfall, carbon availability, biological activity in the soil, and physico-chemical processes, such as volatilization and absorption to the surfaces, are important at various times of the year. We also

discovered that understorey vegetation is a strong sink of monoterpenes. Our statistical model, based on prevailing temperature, seasonality, vegetation effect, and the interaction of these parameters, explained 43% of the monoterpene fluxes, and 34–46% of individual α-pinene, camphene, β-pinene and $\Delta^3$-carene fluxes.

## 1 Introduction

Vegetation in coniferous forests is a primary and well-quantified source of biogenic volatile organic compounds (BVOCs) on the shoot level (Rinne et al., 2000; Hakola et al., 2003; Kim et al., 2005; Bäck et al., 2012, Aalto et al., 2014). The boreal forest floor, including tree roots, understorey vegetation (grasses, shrubs, mosses, lichens, and other vegetation) and the organic soil layer (different stages of decomposing litter, a variety of decomposing and other microorganisms) emits isoprenoids

(monoterpenes: Aaltonen et al., 2011, 5 $\mu g\ m^{-2}\ h^{-1}$ and Hellén et al., 2006, 0–373 $\mu g\ m^{-2}\ h^{-1}$, isoprene and sesquiterpenes: Aaltonen et al., 2011, 0.050 and 0.045 $\mu g\ m^{-2}\ h^{-1}$; Hellén et al., 2006, 0–1.9 and 0–0.8 $\mu g\ m^{-2}\ h^{-1}$ β-caryophyllene). Isoprenoids are a lipophilic group of volatile organic compounds (VOCs) emitted in trace amounts. Isoprenoids are poorly water-soluble and highly reactive in the atmosphere.

Forest floor was discovered to be a significant monoterpene source during spring and fall, when photosynthesis is low (Hellén et al., 2006; Aaltonen et al., 2011, 2013). On the forest floor,



understorey vegetation emits monoterpenes (Aaltonen et al., 2011, Faubert et al., 2012) and photosynthesized energy regulates isoprene syntheses (Ghirardo et al., 2011). Large biomass or coverage of understorey vegetation can also decrease the total soil VOC flux because transpiration can induce the formation of water film on the leaf and chamber inner surfaces, which can enhance isoprenoid absorption (Aaltonen et al., 2013). Trees allocate 40–73% of the photosynthesized carbon for root metabolism, growth and root-associated microbes (Grayston et al., 1997), and the largest portion of photosynthesized carbon is consumed in the root-induced respiration of microbes. The belowground carbon allocation of labelled $C^{13}$ from canopy photosynthesis can be 500% higher in August than June (Högberg et al., 2010). Photosynthesized carbon through the roots was shown to currently contribute 54% of soil respiration (Högberg et al., 2001), but 47% of the carbon allocated to roots and mycorrhizal fungi can also be released to the soil microbial metabolism after root death (Fogel and Hunt, 1983). The main monoterpene sources are suggested to degrade litter (Aaltonen et al., 2011; Faiola et al., 2014), while emitted VOCs strongly depend on litter type (Ramirez et al., 2010) and tree roots (Lin et al., 2007; Aaltonen et al., 2011, 2013), especially damaged ones (Hayward et al., 2001). Forest management can affect the soil isoprenoid fluxes. Clear-cut logging reduced soil VOC fluxes compared to non-disturbed forest soil (Paavolainen et al., 1998), but high monoterpene fluxes are also reported from stumps after a clear-cut (Haapanala et al., 2012). Mycorrhizal fungi also emit oxidized VOCs and small amounts of isoprenoids in a species-specific manner (Bäck et al., 2010). The microbial decomposition of organic matter produces VOCs in soil (Insam and Seewald, 2010; Greenberg et al., 2012). VOCs are often synthesized as side products (aerobic carbon metabolism, fermentation, amino acid degradation, terpenoid biosynthesis and sulfur reduction) during decomposers primary metabolism and energy generation (Peñuelas et al., 2014).

In addition to being released from living or decaying plant material and microorganisms, isoprenoids affect soil processes in multiple ways. Sesquiterpene signalling of mycorrhizal fungi was discovered to enhance root surface area for nutrient uptake and carbon availability for fungi as root exudates (Ditengou et al., 2015). VOCs can induce or reduce microbial activity (Asensio et al., 2012), control the population density of soil organisms (Wenke et al., 2010), and stimulate plant growth as fungal metabolites (Hung et al., 2012). Isoprenoids can inhibit nitrification and mineralization activity by being toxic for some microbes (Smolander et al., 2012), and some bacterial volatiles can have an antagonistic effect on plant pathogens (Kai et al., 2006) or can inhibit or stimulate the growth of soil fungal species (Mackie and Wheatley, 1999). Soil can also be a sink for isoprenoids (Insam and Seewald, 2010, Peñuelas et al., 2014), as some decomposers will also use VOCs as a carbon source (Greenberg et al., 2012). Soil enzymes can release substrates for metabolic VOC production (Mancuso et al., 2015), but isoprenoids can also inhibit enzyme activity in boreal forest soil (Adamczyk et al., 2015).

Soil VOC production processes have not been fully identified in field conditions, despite results showing that they may correspond to tens of percents of the boreal ecosystem flux (Aaltonen et





al., 2013). Microbial VOC production depends on microbial community structure (Bäck et al., 2010), microbial biomass (Wieder et al., 2013), oxygen and nutrient availability (Insam and Seewald, 2010), the physiological state of decomposers (Insam and Seewald, 2010), and substrate quality (Stotzky and Schenk, 1976). Freezing–thawing and drying–wetting events increase isoprenoid fluxes, as they contribute to organic matter degradation (Asensio et al., 2007, 2008; Insam and Seewald, 2010; Aaltonen et al., 2013). Temperature affects VOC production (Asensio et al., 2007), indirectly through the temperature dependence of enzyme production and activity in VOC synthesis (Peñuelas and Staudt, 2010), and directly through volatilization, which is a function of temperature (Guenther et al., 1993). Enclosure techniques are a widely used method to measure soil gas fluxes (Pumpanen et al., 2004), and the enclosure temperature was shown to explain isoprenoid fluxes in stronger way than soil temperature (Hayward et al., 2001, Aaltonen et al., 2013). Increasing temperature and decreasing soil water content contributed higher monoterpene volatilization from soil into the atmosphere (van Roon et al., 2005). Soil water content can also determine which microbial groups are most active (Veres et al., 2014). The flux rate depends on the compound. Monoterpenes are released from storage structures when temperature-dependent vapor pressure changes (Schurgers et al., 2009). High isoprenoid fluxes are also measured after rain events (Greenberg et al., 2012).

This experiment was designed to determine whether carbon allocation to soil via roots affects soil isoprenoid fluxes through root metabolism and microbial activity, when the trenching approach was assumed to change the microbial communities between the different treatments. The aim was to identify isoprenoid sources, quantify isoprenoid fluxes and estimate the parameters regulating the isoprenoid fluxes based on the following hypotheses: (1) Presence of roots and mycorrhizal fungi enhances the amount of structurally non-bound (labile, e.g. fast turnover rate) carbon in the soil, which will increase isoprenoid fluxes. (2) Understorey vegetation is a sink of isoprenoids, as isoprenoids can be adsorbed on leaf surfaces. (3) A statistical model including prevailing temperature, seasonality, trenching treatments, understorey vegetation cover, above-canopy PAR, soil water content, and soil temperature to estimate isoprenoid fluxes.

## 2 Material and methods

### 2.1 Trenching experiment

Measurements were executed in the southern boreal forest at the SMEAR II (Station for Measuring Ecosystem-Atmosphere Relations) station (61º51'N, 24º17'E, 180 m above sea level) (Hari and Kulmala, 2005). The forest is a 55yr old Scots pine stand (*Pinus sylvestris*), where *Sorbus aucuparia*, *Betula pendula* and *Picea abies* grow below-canopy. Soil above the bedrock is Haplic podzol and soil depth is approximately 0.5–0.7 m. The average thickness of the soil horizons from the SMEAR II stand is 6.0 cm (organic layer), 2.0 cm (E-horizon) and 16 cm (B-horizon). The stand was established by sowing after prescribed burning in 1962. Current canopy height is ca. 17 m and one-side leaf area index





(LAI) is 2.0–2.5 $m^2\ m^{-2}$ (Aalto et al., 2014). The understorey vegetation is formed by shrubs, such as *Vaccinium vitis-idaea, Vaccinium myrtillus,* and *Calluna vulgaris,* mosses, such as *Pleurozium schreberi, Dicranum polysetum, Dicranum scorparium,* and *Hylocomium splendens* and grasses such as *Deschampsia flexuosa* and *Melampyrum sylvaticum*. Soil surface coverages of the different vascular and moss species on the experimental plots were determined using the eye estimation method in July 2015. Measurements were conducted on three replicate experimental sites (1, 2, and 3) at the station. Site 1 is directed towards the east, site 2 towards the south-east and site 3 towards the south-east. The distance between replicate sites was 50–100 m. The experimental sites are described in more detail in Table 1.

The experimental setup was established in 2012 to study the effect of carbon allocation by tree roots and mycorrhizal fungi into soil. Each replicate site includes 20 experimental plots with different below- and aboveground treatments, which were implemented to regulate the carbon flow from trees and the understorey vegetation to soil microbes through roots and mycorrhizal fungi. Thirty-six of the experimental plots were measured in our study (Table 2). All the experimental plots were trenched by digging around a square volume (0.9 x 0.9 m) of soil until reaching the bedrock, or to a depth of up to 40 cm, and cutting roots between the experimental plot and the surrounding ground. Soil C input by plant allocation was controlled by comparing the soil, where the ingrowth of roots and mycorrhizal fungi and decomposer mobility was allowed (Control, 18 plots) to experimental plots where the ingrowth of tree roots and fungi was inhibited by placing isolating mesh (1 µm) around the soil volume (code Tr1, 12 plots). The treatment Tr50 (mesh size 50µm, six plots) allowed access for microbes and mycorrhizal fungi, but prevented the ingrowth of tree roots (Table 2). Both meshes allowed water and nutrient exchange. As the understorey vegetation also allocates part of the photosynthetically produced C into the soil, the effect of the understorey vegetation was monitored by comparing plots with different vegetation: either the understorey vegetation was growing normally (marker +, 21 plots), or vegetation was removed by cutting (marker -, 15 plots) (Table 2).

All plots were equipped with a 0.5m long tube, where soil water content was measured using the capacity probe (PR2, Delta-T Devices) every second week. Soil temperature sensors were placed in the soil surface layer (depth 4 cm), and data were logged every fourth hour from May to October in 2012–2015. Photosynthetically active radiation (PAR) was measured from a wavelength range of 400–700 nm using an LI-19OSZ quantum sensor (Li-Cor, Biosciences, Lincoln, NE) at heights of 18.0 m (above canopy) and 0.6 m (below canopy). The monthly total litterfall (needles, bark, twigs, and cones) and fraction of needles in the litterfall was determined once a month at the SMEAR II stand from April to October using 21 litter collectors (diameter 0.48 m).



## 2.2 Measurement methods

The flux of isoprenoids from each plot was measured 5–6 times between 15[th] of April and 23[rd] of October, 2015. To analyze the seasonality of the isoprenoid fluxes, the results were pooled into six periods: 1) 15[th]–24[th] of April, 2) 30[th] of April–10[th] of May, 3) 21[st] of May–24[th] of June, 4) 21st[th] of July

to 21[st] of August, 5) 31[st] of August – 9[th] of October, and 6) 19[th] – 23[rd] of October. The sequence of measurements was randomly arranged, to avoid any systematic errors in flux measurements between plots. The exact timing and sequence of the measurements are presented in the Appendix (Table A1).

Isoprenoid concentrations in the chamber headspace (height 40 cm, chamber volume 10 l) were measured with two dynamic (steady-state flow-through) glass chambers. The chambers were placed on

permanent soil collars (height 7 cm, diameter 21.7 cm), which were placed on each plot in 2012. Incoming and outgoing air was sampled for 1.5–2 hours using sampling flow (0.1 l min[-1]) through two Tenax TA-Carboback-B adsorbent tubes, and the flux was calculated from the difference between ingoing and outgoing air (see Eq1). Filtered (active carbon trap and $MnO_2$-coated copper net) ambient air was continuously pumped (1 l min[-1]) into the chamber, and the chamber air volume was flushed for

0.5 hours before sampling to stabilize the system. Chamber temperature was measured using a thermometer (Fluke 54II, Fluke, WA, USA) from 20–30 cm above ground. Hemiterpenes (isoprene and 2-methyl butenol), monoterpenes (α-pinene, camphene, β-pinene, myrcene, Δ3-carene, p-cymene, limonene, and terpinolene, while oxygen containing 1.8-cineol and linalool are typically categorized for monoterpenes) and sesquiterpenes (longicyclene, iso-longifolene, β-caryophyllene, aromadendrene

and α-humulene) were measured from the adsorbent tubes.

Tenax TA-Carboback-B adsorbent tubes were kept in cold (5 °C) and analyzed the week after sampling using a thermodesorption instrument (Perkin Elmer TurboMatrix 650, Waltham, USA) connected to a gas-chromatograph (Perkin-Elmer Clarus 600, Waltham, USA) and a mass selective detector (Perkin-Elmer Clarus 600&, Waltham, USA). The sample tubes where thermally desorbed for

5 min (300 °C), cryo-focussed in a Tenax cold trap operating at -30 °C, and injected into the column using rapid heating (300 °C) (Aaltonen et al., 2011). Six standards in methanol solutions were used for calibration by injecting into the sample tubes, after which the methanol was flushed away (10 min). The analytical detection limit was 0.005–2.431 µg m[-2] h[-1] for the different isoprenoids.

## 2.3 Flux calculations and statistical analyses

The flux rates ($E$, µg m[-2] h[-1]) of the different compounds were calculated for soil area (area inside to collar, m$^2$) and time ($h$) using Eq. (1):

$$E = \left( C_{out} - C_{in} \right) \frac{F_{chamber}}{1000} \frac{60}{A}, \qquad (1)$$




where $C_{in}$ is the concentration of ingoing air sample (µg m⁻³) and $C_{out}$ is the concentration of outgoing air sample (µg m⁻³), $F_{chamber}$ (m³ min⁻¹) is the flow rate of air pumped into the chamber, and $A$ (m²) is the soil surface area inside the collar.

The detection limit (*DL*) was calculated for every compound and for every measurement week using Eq (2):

$$DL = \left( k_{mean} \left( 3 \sqrt{\frac{\sum(m_{in} - \overline{m_{in}})^2}{n-1}} \right) \right) \frac{F_{chamber\,mean}}{1000} \frac{60}{A},$$  (2)

where $k_{mean}$ is the mean sampled air volume (m³), $c_{in}$ is the compound mass of ingoing air sample (µg

m⁻³), $F_{chamber\,mean}$ (m³ min⁻¹) is the mean flow rate of air pumped into chamber, $A$ (m²) is the soil surface area inside the collar. Data were analyzed using MATLAB software (version 2015a, MathWorks, Natick, MA, USA), and statistical analyses were conducted using SPSS software (version 23, IBM SPSS Statistics; Chicago, IL, USA). R Language and Environment for Statistical Computing program (version 3.2.4; R Core Team, 2016) was used to construct the mixed effects linear models. A random

permuted block design was used in our study with block sizes 3 (Tr50+ and Tr50-), 6 (Control-, Tr1+, Tr1-), and 11 (Control+). The normality of sum monoterpene flux (sum of ten monoterpenes), α-pinene flux, chamber temperature, soil temperature, and soil volumetric water content were tested during six periods using the Kolmogorov-Smirnov and Shapiro-Wilkin tests (degree of freedom: 60=Control+, 28=Control-, 17=Tr50+, 6 =Tr50-, 28=Tr1+, and 24=Tr1-). We also tested whether the annual total

fluxes of different compounds from the trenching treatments were normally distributed. If the data were non-normally distributed, the non-parametric Kruskal-Wallis test (degree of freedom=1) at the significance level of <0.05 was used to determine whether the treatments were statistically different (Table 3 and 4).

The effect of period, vegetation effect, trenching treatment, chamber temperature, above

canopy PAR, soil water content, soil CO₂ flux, and soil temperature for total monoterpene, total sesquiterpene, and individual isoprenoid fluxes were tested using the mixed effects linear models. For example, total monoterpene fluxes (*M*) were modelled by the mixed effects linear model:

$$M = B_0 + B_s + B_v + B_c C + B_{sv} + B_{sc}C + B_{vc}C + B_{svc}C + \epsilon,$$  (3)

where $B_0$ denotes a fixed intercept parameter, $B_s$ are denoting fixed unknown parameters associated with season variable, $B_v$ are denoting fixed unknown parameters associated with vegetation effect variable, $B_c$ are denoting fixed unknown slope parameter related to chamber temperature C, $B_{sv}$ are denoting fixed parameters for interaction of period and vegetation, $B_{sc}$ are denoting fixed slope parameters for interaction of season and chamber temperature, $B_{vc}$ are denoting fixed slope parameters

parameters for interaction of season and chamber temperature, $B_{vc}$ are denoting fixed slope parameters





for interaction of vegetation and chamber temperature, $B_{svc}$ are denoting fixed parameters for three way interaction of period, vegetation, and chamber temperature. In the model (3), the error term $\in$ is assumed to have a form:

$$\in\,=\,\propto_l + \propto_p + u, \qquad\qquad\qquad (4)$$

where $\propto_l$ are denoting random parameters related trenching plot, $\propto_p$ are denoting random parameters related to the measurement site (1, 2, and 3), and $u$ is unobservable random error term. Random effects parameters and random error term are assumed to follow normal distributions $\propto_l\, \sim N(0, \sigma_l^2), \propto_p\, \sim N(0, \sigma_p^2),$ and $u \sim N(0, \sigma_u^2),$ respectively.

Similar type of mixed models with different variable combinations (factor variables are period, vegetation effect, trenching treatment, and numerical variables are chamber temperature, above canopy PAR, soil water content, soil $CO_2$ flux, and soil temperature) were used to model total sesquiterpene fluxes and individual isoprenoid fluxes (Table 5 and Table 6).

## 3 Results

### 3.1 Correlations of temperature, soil moisture, and PAR with VOC measurements

At the experimental sites, soil water content and chamber and soil temperature were measured to observe their influence on the fluxes. Ambient air temperature and PAR were measured at the SMEAR II stand. Daily temperature followed PAR (Fig. 1a). Chamber, soil and ambient temperatures followed very similar patterns. The median difference between chamber and soil temperature was 3.6 °C, and the median difference between chamber and ambient temperature was 0.9 °C. Water content was 0.06–0.45 $m^3$ $m^{-3}$ in the mineral soil (Fig. 1c), and it was higher from April to end of July and very low from August to the end of October. During the measurements PAR was 10–1440 µmol $m^{-2}$ $s^{-1}$ above the canopy and 1–410 µmol $m^{-2}$ $s^{-1}$ below the canopy (Fig. 1b). Chamber and soil temperature did not differ between treatments, except during July and August (period 4), when soil temperature was higher in Control- (13.5 °C) than in Control+ (12.6 °C). Soil water content was higher in Control+ (0.13 $m^3$ $m^{-3}$) than in Control- (0.10 $m^3$ $m^{-3}$) and higher in Tr1- (0.19 $m^3$ $m^{-3}$) compared to Control+ and Control- in September and early October (period 5) (data not shown).

Temperature dependence of monoterpene and sesquiterpene fluxes were determined by combining all the measurements. Sesquiterpene fluxes showed exponential correlation with chamber temperature ($R^2$=0.26, $p<0.001$, Fig. 2a). Monoterpene fluxes did not correlate with chamber temperature ($R^2$=0.03, $p>0.05$, Fig. 2b).



Monoterpene fluxes from the Tr50-plots (ingrowth of decomposers and mycorrhizal fungi) were higher when chamber temperatures were lower ($R^2$=0.91, p<0.01), but in all other treatments the effects were not significant (p>0.05) (Appendix Fig. A2).

We also analyzed the effects of soil water content and temperature, chamber temperature, PAR, and soil $CO_2$ flux on monoterpene and sesquiterpene fluxes from all the treatments. Although we discovered some statistically significant differences, the $R^2$ values were very small, varying from 0.00 to 0.08 (Appendix Fig. A1).

### 3.2 The effect of understorey vegetation

Understorey vegetation was a monoterpene sink, when isoprenoid fluxes were measured on bare soil or on soil with vegetation cover in different treatments. The sum of the monoterpene fluxes were highest from the bare soil, when the soil was non-trenched (Control) or when decomposers were the only source (Tr1). Instead, the sum of the monoterpene fluxes did not differ between bare soil and soil with vegetation cover, when the ingrowth of mycorrhizal fungi was allowed (Tr50) (Table 3). Sesquiterpene fluxes from various sources were equally low (0.35–0.73 µg m$^{-2}$ h$^{-1}$). The most dominating compounds were α-pinene, camphene, β-pinene, and Δ$^3$-carene, covering 84–94% of the flux spectra (Table 3). The exception was Tr1+, where isoprene covered 20% of the spectrum (Table 3). The most abundant sesquiterpenes emitted were β-caryophyllene and aromadendrene (Table 3).

### 3.3 Different VOC sources in soil

Isoprenoid fluxes were compared between the treatments in the six periods. The mean total monoterpene flux from the treatments was 2.0–78.0 µg m$^{-2}$ h$^{-1}$ and the mean total α-pinene flux was 0.6–60.2 µg m$^{-2}$ h$^{-1}$ (Table 4), with high temporal variation. However, the presence of vegetation and decomposer activity clearly affected the fluxes in July–August (period 4) and October (period 6) (Table 4). In July–August, the presence of vegetation (Control+) significantly decreased the total monoterpene and α-pinene fluxes compared to both Control- and Tr1- (Control-: $p_{monoterpenes}$=0.015 and $p_{α-pinene}$ =0.011; Tr1-: $p_{monoterpenes}$=0.027 and $p_{α-pinene}$ =0.035). In October, the decomposer-only treatment (Tr1-) had significantly higher fluxes than Control+ ($p_{monoterpenes}$=0.027 and $p_{α-pinene}$ =0.027) and Tr1+ ($p_{monoterpenes}$=0.034 and $p_{α-pinene}$ =0.034).

### 3.4 Seasonality of VOC fluxes

Seasonal variations of monoterpene, sesquiterpene, and isoprene fluxes were determined from non-trenched soil with vegetation (Control+). Monoterpene, sesquiterpene, and isoprene fluxes varied between 0–149 µg m$^{-2}$ h$^{-1}$, 0–4 µg m$^{-2}$ h$^{-1}$, and 0–29 µg m$^{-2}$ h$^{-1}$, respectively (Fig. 3). Monoterpene fluxes



were highest in October and lowest in mid-April, but as shown in Fig. A2, they poorly correlated with soil temperature. Soil temperature was close to 0 ºC in early October and between 1 ºC and 5 ºC in mid-April (Fig. 3). Sesquiterpene fluxes were highest in summer. Isoprene fluxes were highest in June and July when temperature and PAR was high (Fig. 3), but interestingly this was also observed in October.

Average total monoterpene fluxes were highest from non-trenched and bare soil in September–October (period 5), and from bare soil with decomposers in October (period 6) (Table 4). As shown in Fig. 4, the flux rates correlate with total litterfall and the fraction of needles in the litterfall. As shown in Fig. 3, the effect of litterfall on monoterpene flux rates occurs after a short delay in October. Monthly total litterfall (8.7–114.2 g m$^{-2}$) and the total amount of needles in the litterfall (1.6–99.7 g m$^{-2}$) varied

at the SMEAR II stand from April to October 2015 (Fig. 4). Monthly total and needle litterfall were 75–92% and 84–98% higher in September, and 58–87% and 66–97% higher in October, respectively, compared to the spring and summer months ($p<0.001$), but total litterfall was also high in July (Fig. 4).

### 3.5 Mixed effects model results

Mixed effects linear models were used to determine, which parameters are best in estimating the flux rates of monoterpenes and sesquiterpenes from boreal forest soil. The best fit was obtained with a combination of several biological and abiotic parameters. The presence of vegetation cover, measurement timing (period) and chamber temperature explained 43% of the individual monoterpene fluxes ($p<0.05$), whereas measurement timing (period) and chamber temperature explained 29% of the

individual sesquiterpene fluxes ($p<0.01$) (Table 5). The effect of the trenching treatment, PAR, soil water content, soil temperature, and soil $CO_2$ flux were also tested, but their effects were non-significant ($p>0.05$).

      Mixed effects linear models were also used to determine, which parameters are best in estimating the flux rates of different individual isoprenoids. When the model included chamber

temperature, vegetation effect, seasonality (period), and the interaction of these parameters, it explained 34–46% of the individual α-pinene, camphene, b-pinene, and Δ3-carene fluxes (Table 6). When the model included seasonality, chamber temperature, soil water content, and the interaction of these, it was able to explain 40% of the variation within the longicyclene fluxes (Table 6). Chamber temperature, seasonality, and the interaction of these, explained 35% of the variation within the α-humulene fluxes.

Seasonality, soil temperature, soil water content, and the interaction of these were able to estimate 35% of the variation within the isoprene fluxes (Table 6).

### 4 Discussion

Identifying the sources of isoprenoid fluxes from forest understorey vegetation and soil in field

conditions is challenging, as most measurement techniques only yield net exchange (including all



sources and sinks). The only way to dissect various processes is to manipulate the system, which was done here during the trenching treatment and vegetation removal. With the presented trenching experiment, where soil biological processes could be separated into different components, it was possible to separately analyze the fluxes originating from the decomposer, mycorrhizal fungal, tree

roots, and understorey vegetation individually for the first time. First, we tested whether the photosynthesized carbon allocation to the soil affects the isoprenoid production of different soil organisms (decomposers, mycorrhizal fungi, and roots). Second, we analyzed how important the vegetation is as a sink. Third, we aimed to construct a statistical model including prevailing temperature, seasonality, trenching treatments, understorey vegetation cover, above-canopy PAR, soil water content,

and soil temperature to estimate isoprenoid fluxes.

### 4.1 Seasonality and carbon source impacts on emission rates and spectra

Our results show that the seasonality of emissions is largely correlated to litterfall, especially for monoterpenes, and our results confirm the emission spectrum and temporal variation of isoprenoids

from the boreal forest understorey and soil layer found by Hellén et al., (2006) and Aaltonen et al., (2011; 2013). Earlier studies have also suggested that litter and decomposers are important isoprenoid sources (Hayward et al., 2001; Asensio et al., 2007, 2008; Isidorov et al., 2010, Insam and Seewald, 2010, Aaltonen et al., 2013, Greenberg et al., 2012, Faiola et al., 2014). Monoterpenes can be produced simultaneously by MEP pathway in plastids and by MVK pathway in cytoplasm, and at least some

fungi and bacteria are capable of activating the MEP pathway (Rohmer et al., 1993; 1996; Eisenreich et al., 1998; Walter et al., 2000; Banerjee and Sharkey, 2014). Soil can also absorb 80% of litter-produced VOCs (Ramirez et al., 2010), when soil and litter samples from a *Pinus taeda* stand on a loamy sand soil (pH 3.6, 50% of the water holding capacity) were studied in a laboratory.

        The litterfall amount reflects the stand density and dominating tree species of the forest canopy,

and indirectly the size of forest carbon storage. VOC release from the fresh litter appears important, as the highest isoprenoid fluxes were measured in October, correlating with litterfall (especially needle) production. Decomposition releases isoprenoids from needle storages (Aaltonen et al., 2011), and litter emissions are regulated by microbial activity i.e. soil respiration, microbial biomass, carbon availability, temperature, and rain events (Leff and Fierer, 2008; Greenberg et al., 2012). Old litter can also be an

important isoprenoid source during the following year, as the degradation of Scots pine litter is a slow process (Kainulainen and Holopainen, 2002). Decomposition can continue in soil under snow cover, and isoprenoids are released after snowmelt (Aaltonen et al., 2011). Isoprenoids can also be released after non-enzymatic, thermo-chemical reactions (Greenberg et al., 2012), and soil processes can be efficient isoprenoid sources also during wintertime (Aaltonen et al., 2012). Litterfall contribution to

decomposition processes is generally significant, as the decomposition of fresh litter requires less



energy than the decomposition of non-labile organic compounds. The quantity of carbon and its decomposability also decreases with litter age (Greenberg et al., 2012).

Contrary to our hypothesis, belowground carbon availability did not clearly affect emissions, as only minor differences were observed between the trenching treatments. This is a significant finding and indicates that despite microbial communities most probably being very different in various trenching treatments, community changes do not significantly affect the net VOC flux from soils. We propose that the reason for this is that VOCs used for microbial signalling (e.g. references, Wenke et al., 2010; Ditengou et al., 2015) are produced in low concentrations and therefore they cannot be seen in the soil net VOC flux. One theory for this would be that the presence of tree roots and plant-derived carbon flow favours microbes that are able to use VOCs as an energy source (Greenberg et al., 2012). However, we were unable to investigate either the microbial community structure or their VOC signalling in our study. As a conclusion, we may say that soil VOC fluxes are likely regulated by other processes than those directly dependent on plant-derived C flow into soil via roots.

In addition to litterfall, seasonal temperature variations also had an effect, especially on the sesquiterpene emissions. This was expected, as temperature can regulate isoprenoid emissions through physical processes (volatility and diffusion) and the enzyme activity of VOC synthesis (Peñuelas and Staudt, 2010). The traditional approach for modelling isoprenoid emissions is to use the so-called Guenther algorithm (Guenther et al., 1991; 1993; 1995), which calculates individual plant- or ecosystem-scale emissions rates according to prevailing temperature. The global emission model MEGAN (Model of Emissions of Gases and Aerosols from Nature) was developed based on the Guenther algorithm, and the model includes plant functional type, long-term temperature response, leaf age, and soil water content (Guenther et al., 2006; 2012). Often this is a good approximation for forest ecosystem- or global-scale inventories of biogenic VOC emissions (Grote and Niinemets, 2008; Sindelarova et al., 2014; Chatani et al., 2015). However, the effects of temperature on emissions from soil are not straightforward, as the soil biological activity is very different between spring and autumn, although air or soil temperatures may be very similar. This was clearly seen in our results. Sesquiterpenes are known to be signalling compounds between the roots and ectomycorrhizal fungi (Ditengou et al., 2015), and this signalling could be stronger during active periods of the tree. Sesquiterpene flux rates were small in our study, possibly as they can react in the topsoil, or on leaf surfaces before they are released into the chamber headspace. Sesquiterpene volatilization also requires a higher temperature than the volatilization of monoterpenes, and the adsorption of sesquiterpenes on leaf and chamber surfaces is more likely than monoterpene adsorption. Other effects of carbon availability on isoprenoid fluxes were not confirmed.



### 4.2 Effect of understorey vegetation on VOC fluxes

The most important contributing factor to net flux from the forest floor during the entire growing period seems to be the vegetation cover, which was discovered to be a sink for isoprenoids. The difference in total monoterpene fluxes between the vegetated and bare soil plots was largest in July–August (soil with decomposers only, 8.5-fold) and in mid-October (non-trenched soil, 3.5-fold), and the average flux difference between the two treatments was 2.8-fold. Isoprenoids, especially monoterpenes, were likely absorbed on the leaf surfaces. Leaf surfaces are covered by a lipophilic cuticle layer that offers protection against environmental stress (cold, UV light, drought etc.) (Pollard et al., 2008). Monoterpenes, as lipophilic and volatile compounds, can be absorbed on the lipophilic cuticle layer (Joensuu et al., 2016). The lowest isoprenoid fluxes were previously measured from soil with dense understorey vegetation cover (Aaltonen et al., 2013), which supports our conclusion.

The *Vaccinium spp.* -dominated understorey vegetation in Scots pine forests also synthesize monoterpenes (Faubert et al., 2012). Janson et al., (1999) and Aaltonen et al., (2011) reported isoprenoid emissions from a forest floor covered with shrubs, such as *Vaccinium myrtillus*, mosses, such as *Pleurozium schreberi and Hylocomium splendens,* and grasses such as *Melampyrum sylvaticum*. Kesselmeier et al., (1999) reported that *Pleurozium schreberi* emits aldehydes. Temperate grassland species have been observed to emit isoprenoids (He et al., 2005), along with Mediterranean plant species (Owen et al., 2001), crop species, and tree species (Karl et al., 2009; Laothawornkitkul et al., 2009) such as *Betula nana, Salix sp., Cassiope tetragona, Populus tremula* (Hakola et al., 1998; Rinnan et al., 2011). Hewitt and Street (1992) and Rinnan et al., (2014) discovered that *Deschampsia sp.* does not emit isoprene or monoterpenes. Subarctic heath emits isoprenoids (Faubert et al., 2012). Mosses are important to consider in the forest floor VOC exchange, as they emit isoprene (Hanson et al., 1999) and produce up to 40% of the gross photosynthetic production of the understorey vegetation at the SMEAR stand (Kolari et al., 2006).

### 4.3 Testing the factors involved in VOC flux from the forest floor

This experimental setup was designed to determine whether carbon allocation to soil via the roots affects soil isoprenoid fluxes through root metabolism and microbial activity, and whether radiation-driven photosynthesized carbon availability for roots and microbes regulates isoprenoid fluxes. According to our statistical model, belowground carbon availability does not significantly affect the boreal forest soil isoprenoid exchange.

Our measurement setup enables us to test contributing factors for isoprenoid emissions by constructing a statistical model. Different statistical models were tested, but only the parameters with a statistically significant effect were included, and the best model with the highest explanatory power was chosen. The best model included seasonality, vegetation effect, prevailing temperature, and the





interaction of these parameters was able to explain 29–43% of the variation within monoterpene and sesquiterpene fluxes, which means that a significant portion of the variation was solved. We were also able to construct a model explaining 46% of the individual α-pinene fluxes based on vegetation effect, seasonality, prevailing temperature, and the interaction of these parameters. This indicates that separate models should always be built for different compound groups (monoterpenes and sesquiterpenes) with different physical and chemical properties.

We were able to estimate the flux rates with decent explanation power, although more improvement should be achieved in the future. Possible reasons behind the unsolved emission rates are oxygen and nutrient availability (Rinnan et al., 2011, the fertilization effect of *Salix phylicifolia* on the β-selinene flux*)*, quality and quantity of the organic matter, soil composition, and microbial community structure, which were not determined in our study. It is also possible that some tested parameters were non-linear, and for this reason were unsuitable parameters into the mixed effects linear model. A process-based model should be built in the future, as it would increase our understanding of the forest floor isoprenoid exchange by including dependencies of the different environmental parameters and soil processes.

### 4.4 Error sources in the measurements

Isoprenoids are difficult to measure under field conditions, as they are emitted in trace amounts and are highly reactive, which means that they can disappear through chemical reactions before they have been sampled or analyzed. Sesquiterpenes, with very small emission rates, low volatility, and high reactivity, are especially difficult to detect and quantify (Guenther et al., 2013). Sesquiterpene flux rates are probably underestimated more than isoprene and monoterpene flux rates, since daytime reactivity (OH and $O_3$) is 1.3 min for β-caryophyllene, 27 min for isoprene, 29 min for $\Delta^3$-carene, and 41 min for α-pinene (Rinne et al., 2007), although the majority of oxidants are filtered before the chamber headspace. The difference in emission rates between treatments can be smaller than the random errors in the measured fluxes, produced by the sampling and analysis system.

As the sampling time should to be considerably long (here: 1.5–2 hours) to exceed the detection limit of the TD-GC-MS, this means that the results are cumulative emissions over the sampling time. The chamber method combined with adsorbent sampling thus measures the net exchange with simultaneous sources and sinks, and this can only be overcome with fast-response analytical methods such as Proton-transfer reaction Mass spectrometry (PTR-MS). However, the speciation to different compounds is only possible with a TD-GC-MS, which is why we chose to use this method.

Temperatures inside the enclosure typically increase during the measurements, especially if the enclosure time is long and the chamber is in direct sunlight. This can cause overestimations in the flux rates, when increasing temperature affects the volatility and diffusion rate of the compounds (Niinemets et al., 2011). Luckily, in our study, the median difference of enclosure



temperature and ambient air was small (0.9 °C) for the entire data set, probably since we used a fan inside the chamber and the purge flow rate was approximately 1 lpm. Further, temperatures close to the soil surface are rather stable due to the lack of direct sunlight under the closed canopy.

Many isoprenoids are released in large amounts from cut surfaces or due to the rough handling of measured plants. The trenching and cutting of vegetation was performed three years prior to these measurements. Since the distance from our measurement collar to the closest trench was 30 cm, we assume that the effect of root cutting is very small. Mechanical removal (cutting) of vegetation could cause some local effects and random variation to the plots where the vegetation was removed, but since the need for repeated cutting in the third year was rather small, and it was mostly performed in spring, we believe that it did not significantly affect the fluxes later on. Soil surfaces in the cut treatments were still partly covered by mosses (16–20%), as it is impossible to remove a very thin moss cover without disturbing the organic soil. This may influence the observed differences between bare soil and soil with vegetation, as mosses are known to emit isoprene (Hanson et al., 1999). A minor trend was observed where the highest isoprene emissions occurred when the fraction of mosses made up over 55% of the soil surface coverage.

Soil is a highly heterogenic matrix, where soil depth, nutrient status, root density, and water content can vary based on vegetation cover, shading and soil composition (porosity, texture, and stoniness). High spatial and temporal variation can make differences between the treatments more difficult to detect.

## 5 Conclusions

Our results show that belowground carbon availability does not play a major role in isoprenoid exchange, but instead the litterfall, i.e. carbon from above, is important. Our results emphasize that the net sink effect of understorey vegetation should be included for modelling forest VOC exchange. These results add to our knowledge concerning forest floor VOC fluxes for modelling stand-level VOC exchange. The accurate quantification of soil VOC fluxes can improve air chemistry models, where the difference in the hydroxyl radical (OH) reactivity sink between the measurements and air chemistry models is most likely due to the unknown VOC sources (Mogensen et al., 2011). OH is the most important oxidant for atmospheric VOCs, and more accurate quantification of the OH reactivity sink is needed to enhance our understanding of the atmospheric capacity to oxidize gas-phase organic trace gases for secondary organic aerosol formation (SOA).

*Author contribution.* Manuscript preparation and analyzing results (M.M.). All authors contributed to project planning, experimental design, the discussion of the results and commenting on the manuscript.



*Acknowledgements*. This research was founded by the Academy of Finland Center of Excellence programme (grant no 272041) and by the Jenny and Antti Wihuri Foundation. The research was also supported by the Academy research fellow projects (Academy of Finland, projects 275608, 263858, and 292699). We also thank Eeva-Stiina Tuittila and Aino Korrensalo for the understorey vegetation

survey on the experimental plots, Jarkko Isotalo for the guidance on the statistical analysis and the staff of the SMEAR II station and Hyytiälä Forestry Field station.

*Disclaimer*: The authors declare that the work is original contribution not submitted elsewhere, and there are no competing financial interests.

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



## Tables and Figures

**Table 1.** Soil depth (cm) and soil surface coverages (%) of mosses, ericoid shrubs, grasses, and tree seedlings on the soil-vegetation interface (+) and on bare soil, where vegetation was removed by cutting (-) on all experimental plots at three experimental sites (1, 2, and 3) in 2015. The standard error of the mean is given next to the mean.

| Site | Vegetation | Soil depth | Mosses | Ericoid shrubs | Grasses | Tree seedlings |
|------|-----------|-----------|-----------|----------------|-------------|----------------|
| 1 | + | 41.1 (5.4) | 67.8 (9.7) | 35.4 (9.1) | 8.4 (5.5) | 0.2 (0.2) |
|   | - | 45.3 (3.3) | 20.1 (6.5) | 0.2 (0.1) | 0.0 (0.0) | 0.0 (0.0) |
| 2 | + | 34.1 (4.3) | 69.9 (9.9) | 30.4 (6.4) | 17.5 (12.4) | 0.1 (0.1) |
|   | - | 46.4 (5.5) | 17.1 (2.4) | 0.0 (0.0) | 0.1 (0.0) | 0.0 (0.0) |
| 3 | + | 41.8 (7.4) | 67.7 (8.7) | 24.1 (6.6) | 8.7 (5.7) | 0.4 (0.3) |
|   | - | 43.7 (3.8) | 16.1 (7.5) | 2.4 (2.1) | 0.1 (0.0) | 0.0 (0.0) |

**Table 2.** Number (N) of measured experimental plots on the different trenching treatments (Control: soil was non-trenched, Tr50: the ingrowth of mycorrhizal fungi was allowed, and Tr1: decomposers were the only source) with vegetation (+) and those with bare soil (-) and the total number of plots.

| Treatment | N | Treatment | N | Treatment | N |
|-----------|----|-----------|---|-----------|---|
| Control+ | 12 | Tr50+ | 3 | Tr1+ | 6 |
| Control- | 6 | Tr50- | 3 | Tr1- | 6 |
|          |    | Total | 36 |       |   |





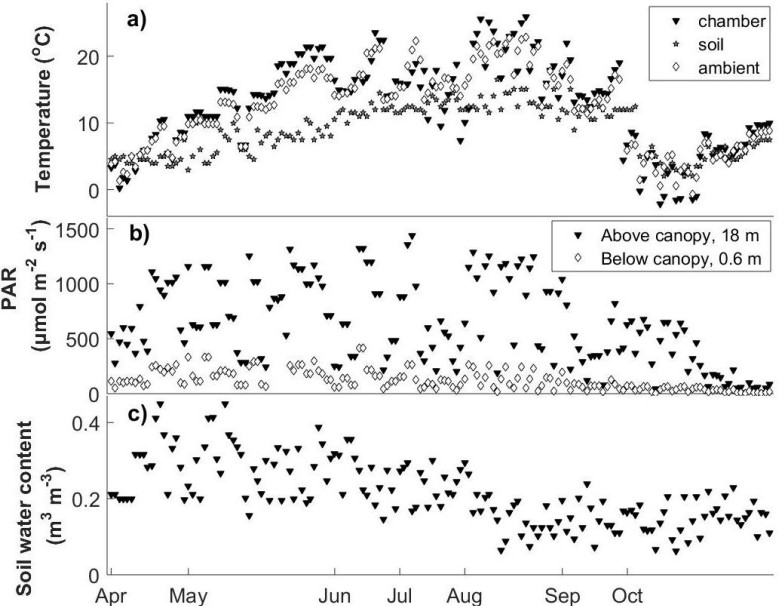

**Figure 1.** Environmental parameters during the measurements from 15th of April to 23th of October 2015. a) Chamber, soil, and ambient temperature (°C). b) Photosynthetically active radiation (PAR, μmol m$^{-2}$ s$^{-1}$) above and below the canopy. c) Soil water content (m$^3$ m$^{-3}$).

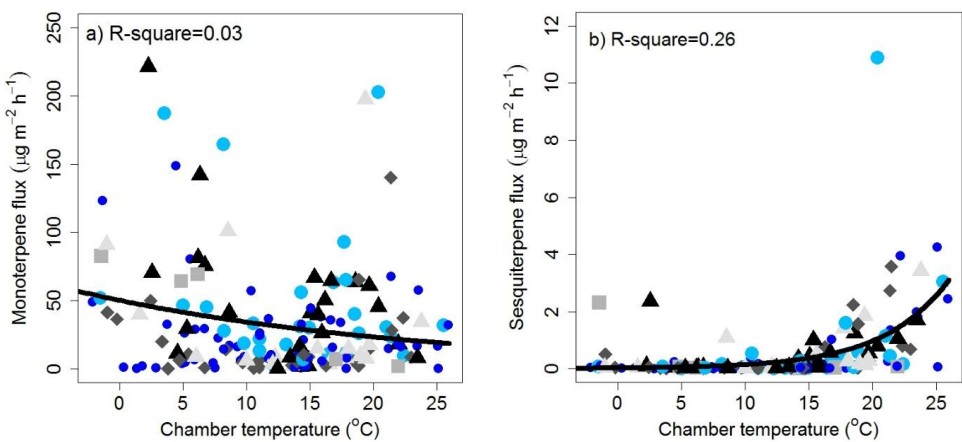

**Figure 2.** Relationship between monoterpene (a, $R^2$=0.03, p-value<0.01) and sesquiterpene (b, $R^2$=0.26, p-value>0.05) fluxes (μg m$^{-2}$ h$^{-1}$) and chamber temperature, presented as combined data from the different treatments (Control: soil was non-trenched, Tr50: the ingrowth of mycorrhizal fungi was allowed, and Tr1: decomposers were the only source) with vegetation (+) and those with bare soil (-). The treatments (Control+=small blue circle, Control-=solid blue circle, Tr50+=filled gray triangle





point-up, Tr50-=filled gray square, Tr1+=filled gray diamond, and Tr1-=filled black triangle point-up) were measured during periods 1–6, 2015.

**Table 3.** Isoprenoid fluxes (µg m$^{-2}$ h$^{-1}$) from the different trenching treatments (Control+ (N=60), Control- (N=28), Tr1+ (N=28), Tr1- (N=24), Tr50+ (N=17) and Tr50- (N=6)) during periods 1–6, 2015. Fluxes are means (S.E.) of the whole data set. BDL = below detection limit. The effect of vegetation on fluxes between the plots with vegetation (+) and those with bare soil (-) was tested with the Kruskal-Wallis test (p<0.05). Values were marked in bold if they differed between vegetation treatments. Significant differences in flux rates between the trenching treatments are indicated with different letters (Kruskal-Wallis test; p<0.05).

| Flux | Control+ | Control- | Tr50+ | Tr50- | Tr1+ | Tr1- |
|---|---|---|---|---|---|---|
| isoprene | 1.60[a] (0.56) | 0.98[a] (0.33) | 4.43[a] (2.78) | 1.24[a] (0.93) | 4.91[a] (3.55) | 4.10[a] (2.27) |
| **Monoterpenes** | | | | | | |
| α-pinene | **14.68[a] (2.57)** | **31.35[b] (6.93)** | 21.98[ac] (8.41) | 26.35[ab] (11.15) | **11.53[a] (3.42)** | **36.18[bc] (8.09)** |
| camphene | **1.70[ac] (0.23)** | **4.34[b] (1.05)** | 2.87[ab] (0.87) | 3.08[abc] (0.87) | **1.39[c] (0.39)** | **3.07[b] (0.55)** |
| β-pinene | **0.30[a] (0.06)** | **0.70[b] (0.18)** | 0.46[ab] (0.21) | 0.44[ab] (0.24) | **0.25[a] (0.08)** | **0.53[ab] (0.14)** |
| myrcene | **0.09[a] (0.02)** | **0.21[b] (0.06)** | 0.19[ab] (0.10) | 0.18[ab] (0.09) | **0.14[ab] (0.04)** | **0.23[b] (0.06)** |
| Δ3-carene | **5.41[a] (0.79)** | **10.97[b] (2.17)** | 7.32[ac] (2.87) | 7.83[ab] (3.09) | **5.25[a] (1.64)** | **8.57[bc] (1.44)** |
| p-cymene | **0.19[ad] (0.07)** | **0.29[b] (0.05)** | 0.16[ac] (0.06) | 0.13[abc] (0.04) | **0.13[c] (0.05)** | **0.23[bd] (0.04)** |
| limonene | **0.29[a] (0.05)** | **0.49[b] (0.09)** | 0.34[ac] (0.15) | 0.23[ab] (0.10) | **0.27[a] (0.09)** | **0.44[bc] (0.07)** |
| terpinolene | **0.05[a] (0.01)** | **0.09[b] (0.03)** | 0.09[ab] (0.04) | 0.07[ab] (0.03) | **0.05[a] (0.02)** | **0.09[b] (0.02)** |
| **Sum of the monoterpenes** | **22.87[a]** | **48.62[b]** | 33.59[ac] | 38.43[ab] | **19.18[a]** | **49.49[bc]** |
| **Sesquiterpenes** | | | | | | |
| longicyclene | 0.01[a] (0.002) | 0.01[a] (0.002) | 0.01[a] (0.004) | BDL | 0.01[a] (0.002) | 0.01[a] (0.002) |
| β-caryophyllene | 0.24[a] (0.073) | 0.51[a] (0.273) | 0.39[a] (0.150) | 0.34[a] (0.317) | 0.38[a] (0.140) | 0.34[a] (0.106) |
| aromadendrene | 0.07[a] (0.026) | 0.16[a] (0.093) | 0.10[a] (0.052) | BDL | 0.06[a] (0.023) | 0.07[a] (0.023) |
| α-humulene | 0.03[a] (0.010) | 0.06[a] (0.027) | 0.05[a] (0.022) | 0.06[a] (0.062) | 0.05[a] (0.021) | 0.03[a] (0.010) |
| **Sum of the sesquiterpenes** | 0.35[a] | 0.73[a] | 0.55[a] | 0.42[a] | 0.50[a] | 0.45[a] |

**Table 4.** Mean (S.E.) total monoterpene and α-pinene fluxes (µg m$^{-2}$ h$^{-1}$) from different treatments (Control+, Control-, Tr50+, Tr50-, Tr1+ and Tr1-) during periods 1 to 6, 2015. The periods are 1) 15$^{th}$–24$^{th}$ of April, 2) 30$^{th}$ of April to 10$^{th}$ of May, 3) 21$^{st}$ of May to 24$^{th}$ of June, 4) 21$^{st}$ of July to 21$^{st}$ of August, 5) 31st of August to 9$^{th}$ of October and 6) 19$^{th}$–23$^{rd}$ of October. Values were denoted different letters (a, b, and c) if they differed between treatments within the certain time period (Kruskal-Wallis test; p<0.05).

| Period | Control+ | Control- | Tr50+ | Tr50- | Tr1+ | Tr1- |
|---|---|---|---|---|---|---|
| | | | **Monoterpenes** | | | |
| 1 | 13.0 (5.8) | 30.5 (2.7) | - | - | 10.2 (9.8) | - |
| 2 | 17.9 (5.5) | 16.6 (6.2) | 37.5 (31.7) | 5.4 (-) | 5.1 (1.8) | 18.1 (11.8) |
| 3 | 21.4 (6.9) | 52.4 (25.9) | 58.4 (46.5) | - | 34.4 (21.8) | 39.9 (10.2) |
| 4 | 13.8[a] (3.4) | 48.0[b] (11.9) | 15.4 (9.6) | 2.0 (-) | 18.6 (10.2) | 39.5[b] (11.8) |
| 5 | 44.8 (12.8) | 78.0 (31.7) | 32.7 (15.7) | 51.3 (22.8) | 24.5 (8.4) | 67.1 (33.4) |
| 6 | 19.3[a] (3.8) | 36.9 (9.0) | 7.6 (1.5) | 69.2 (-) | 8.6[b] (2.9) | 73.5[c] (25.4) |





| | | | α-pinene | | | |
|---|---|---|---|---|---|---|
| 1 | 7.0 (3.1) | 16.8 (1.0) | - | - | 7.9 (7.6) | - |
| 2 | 10.7 (3.5) | 6.7 (2.8) | 20.4 (19.4) | 0.6 (-) | 2.1 (0.9) | 11.5 (7.6) |
| 3 | 13.0 (4.3) | 31.7 (16.6) | 38.3 (31.1) | - | 21.1 (13.5) | 25.3 (7.3) |
| 4 | 7.6[a] (2.2) | 30.9[b] (8.9) | 9.2 (6.8) | 0.8 (-) | 7.9[a] (4.3) | 28.2[b] (9.6) |
| 5 | 31.8 (9.4) | 55.1 (22.9) | 24.3 (12.2) | 35.7 (16.4) | 17.7 (6.4) | 50.0 (25.6) |
| 6 | 13.2[a] (2.6) | 26.5 (6.9) | 5.3 (1.0) | 49.6 (-) | 5.4[a] (1.7) | 60.2[b] (22.8) |

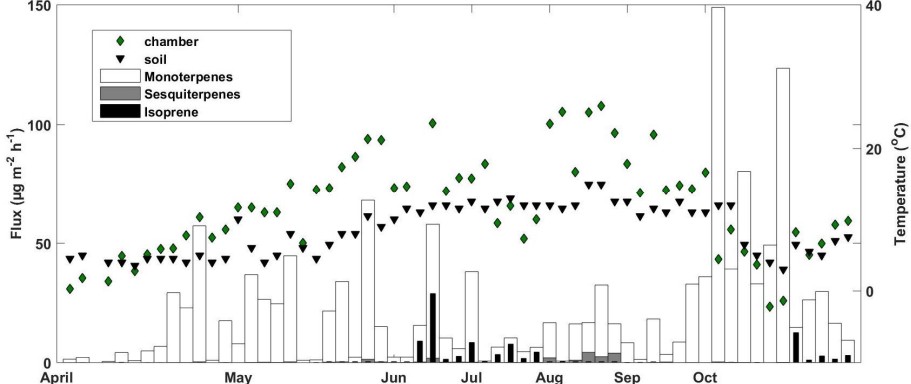

**Figure 3.** Monoterpene, sesquiterpene, and isoprene fluxes ($\mu g\ m^{-2}\ h^{-1}$), and chamber and soil temperatures ($^{o}C$) from a non-trenched forest floor (Control+) during April–October 2015.

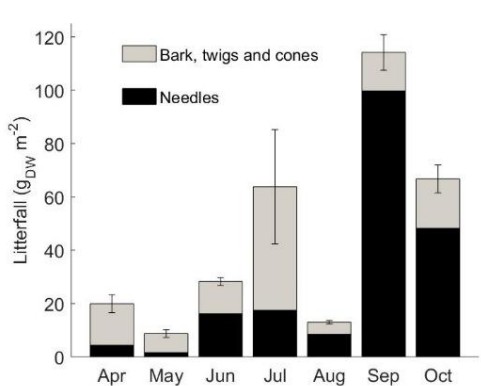

**Figure 4.** Monthly litterfall of bark, twigs, and cones (grey, g (DW) $m^{-2}$), and the fraction of needles in litterfall (black, g (DW) $m^{-2}$) at the SMEAR II stand from April to October 2015. Error bars indicate the standard error of monthly total litterfall from 21 litter collectors.



**Table 5.** Results from the mixed effects linear models, testing the factors impacting monoterpene and sesquiterpene fluxes from boreal forest soil (N [all treatments] = 163, N [plot] = 36, N [site] = 3). Tested effects: period (1–6), vegetation (+/-), chamber temperature (chamber temp), soil temperature (soil temp), PAR, soil water content (soil wt), and the interactions of these. Random effects were related to trenching plot number and trenching site. Pseudo-R-squared was calculated based on Nakagawa and Schielzeth, (2013), and Johnson et al., (2014).

**MONOTERPENES**

| Fixed effects: | Chisq value | p-value | Pseudo-R-squared |
|---|---|---|---|
| factor (period) | 16.762 | 0.004975 ** | 0.43 |
| factor (vegetation) | 12.52 | 0.0004026 *** | |
| chamber temp | 7.7944 | 0.005241 ** | |
| period*vegetation | 6.9411 | 0.2251 | |
| period*chamber temp | 27.771 | 4.035e-05 *** | |
| vegetation*chamber temp | 4.5996 | 0.03198 * | |
| period*vegetation*chamber temp | 5.3451 | 0.3752 | |

**SESQUITERPENES**

| Fixed effects: | Chisq value | p-value | Pseudo-R-squared |
|---|---|---|---|
| factor (period) | 7.0716 | 0.2154 | 0.29 |
| chamber temp | 6.8436 | 0.008896 ** | |
| period*chamber temp | 22.44 | 0.0004318 *** | |





**Table 6.** Results from the mixed effects linear models, testing the factors impacting isoprenoid fluxes from boreal forest soil (N [all treatments] = 163, N [plot] = 36, N [site] = 3). Tested effects: period (1-6), vegetation (+/-), chamber temperature (chamber temp), soil temperature (soil temp), PAR, soil water content (soil wt), and the interactions of these. Random effects were related to trenching plot number and trenching site. Pseudo-R-squared was calculated based on Nakagawa and Schielzeth, (2013), and Johnson et al., (2014).

| Fixed effects: | Chisq value | p-value | Pseudo-R-squared | Chisq value | p-value | Pseudo-R-squared |
|---|---|---|---|---|---|---|
| **MONOTERPENES** | | | | | | |
| | | **α-pinene** | | | **camphene** | |
| factor (period) | 20.206 | 0.001143 ** | 0.46 | 10.281 | 0.06764 | 0.44 |
| factor (vegetation) | 13.086 | 0.0002975 *** | | 11.928 | 0.000553 *** | |
| chamber temp | 11.28 | 0.0007833 *** | | 0.8389 | 0.3597 | |
| period*vegetation | 8.7498 | 0.1195 | | 1.1673 | 0.948 | |
| period*chamber temp | 25.809 | 9.717e-05 *** | | 28.527 | 2.87e-05 *** | |
| vegetation*chamber temp | 5.4705 | 0.01934 * | | 1.0471 | 0.3062 | |
| period*vegetation*chamber temp | | 0.3903 | | 11.508 | 0.04219 * | |
| | | **b-pinene** | | | **Δ3-carene** | |
| factor (period) | 25.781 | 9.841e-05 *** | 0.39 | 9.6409 | 0.08607 | 0.34 |
| factor (vegetation) | 7.8661 | 0.005037 ** | | 7.169 | 0.007417 ** | |
| chamber temp | 6.6896 | 0.009697 ** | | 1.7575 | 0.1849 | |
| period*vegetation | 7.0668 | 0.2157 | | 5.0279 | 0.4125 | |
| period*chamber temp | 21.477 | 0.000658 *** | | 27.831 | 3.927e-05 *** | |
| vegetation*chamber temp | 3.2511 | 0.07138 | | 2.6246 | 0.1052 | |
| period*vegetation*chamber temp | 6.391 | 0.27 | | 3.0477 | 0.6926 | |
| | | **limonene** | | | | |
| factor (period) | 11.947 | 0.03552 * | 0.38 | | | |
| PAR | 5.407 | 0.02006 * | | | | |
| chamber temp | 0.2088 | 0.6477 | | | | |
| period*PAR | 12.302 | 0.03087 * | | | | |
| period*chamber temp | 5.542 | 0.3534 | | | | |
| PAR*chamber temp | 5.9393 | 0.01481 * | | | | |
| period*PAR*chamber temp | 8.4248 | 0.1343 | | | | |
| **SESQUITERPENES** | | | | | | |
| | | **longicyclene** | | | | |
| factor (period) | 13.364 | 0.0202 * | 0.40 | | | |
| soil wt | 4.2641 | 0.03893 * | | | | |
| chamber temp | 8.8191 | 0.002981 ** | | | | |
| period*soil wt | 0.8172 | 0.9759 | | | | |
| period*chamber temp | 21.212 | 0.0007388 *** | | | | |
| soil wt*chamber temp | 5.403 | 0.0201 * | | | | |
| period*soil wt*chamber temp | 9.9874 | 0.07559 | | | | |
| | | **α-humulene** | | | **β-caryophyllene** | |
| factor (period) | 11.38 | 0.04434 * | 0.35 | 5.9382 | 0.3123 | 0.31 |
| chamber temp | 3.5212 | 0.06059 | | 6.0838 | 0.01364 * | |
| period*chamber temp | 22.849 | 0.0003608 *** | | 21.981 | 0.0005279 *** | |
| **ISOPRENE** | | | | | | |
| factor (period) | 5.5947 | 0.3477 | 0.35 | | | |
| soil wt | 1.077 | 0.2994 | | | | |
| soil temp | 5.4103 | 0.02002 * | | | | |



| | | |
|---|---|---|
| period*soil wt | 10.32 | 0.06665 |
| period*soil temp | 25.991 | 8.958e-05 *** |
| soil wt*soil temp | 0.1811 | 0.6705 |
| period*soil wt*soil temp | 15.851 | 0.007282 ** |

**Appendix**

**Table A1.** Chamber measurements from the different trenching treatments (Control+, Control-, Tr1+, Tr1-, Tr50+, Tr50-) from 15th of April to 23th of October 2015.

| Time | Trt | Plot | Time | Trt | Plot | Time | Trt | Plot |
|---|---|---|---|---|---|---|---|---|
| 15th April 1 PM | Tr1+ | 2 | 23rd May 4 PM | Tr50+ | 19 | 31st August 1 PM | Control+ | 37 |
| 15th April 3 PM | Tr1+ | 1 | 22nd June 10 AM | Control- | 52 | 31st August 4 PM | Control- | 43 |
| 16th April 10 AM | Control+ | 38 | 22nd June 10 AM | Tr1+ | 14 | 1st September 8 AM | Control+ | 39 |
| 16th April 1 PM | Control+ | 38 | 22nd June 12 AM | Tr1- | 18 | 1st September 11 AM | Tr50- | 11 |
| 17th April 9 AM | Control+ | 38 | 22nd June 1 PM | Control+ | 46 | 1st September 1 PM | Control+ | 38 |
| 17th April 1 PM | Control+ | 38 | 22nd June 3 PM | Control+ | 48 | 1st September 4 PM | Tr50+ | 7 |
| 23rd April 10 AM | Control+ | 53 | 22nd June 3 PM | Tr50+ | 19 | 2nd September 9 AM | Control- | 44 |
| 23rd April 12 PM | Control+ | 53 | 23rd June 9 AM | Control+ | 53 | 2nd September 11 AM | Control+ | 40 |
| 23rd April 2 PM | Control+ | 53 | 23rd June 10 AM | Tr1- | 29 | 2nd September 2 PM | Tr1+ | 2 |
| 24th April 3 PM | Control+ | 37 | 23rd June 12 PM | Control- | 59 | 2nd September 4 PM | Tr1- | 6 |
| 25th April 9 AM | Control+ | 56 | 23rd June 12 PM | Tr50+ | 31 | 3rd September 9 AM | Tr1- | 17 |
| 25th April 10 AM | Control- | 51 | 23rd June 3 P M | Control+ | 55 | 3rd September 11 AM | Control- | 51 |
| 25th April 12 PM | Control+ | 48 | 23rd June 3 PM | Tr1+ | 25 | 3rd September 1 PM | Control+ | 45 |
| 25th April 1 PM | Control- | 52 | 24th June 8 AM | Control- | 60 | 3rd September 4 PM | Tr1+ | 13 |
| 30th April 9 AM | Tr1+ | 1 | 24th June 8 AM | Control+ | 54 | 4th September 9 AM | Control+ | 47 |
| 30th April 9 AM | Tr1- | 5 | 24th June 11 AM | Tr1- | 30 | 4th September 11 AM | Control+ | 47 |
| 30th April 11 AM | Control+ | 39 | 24th June 11 AM | Tr1+ | 26 | 4th September 1 PM | Tr50+ | 19 |
| 30th April 2 PM | Control+ | 37 | 24th June 1 PM | Control+ | 56 | 4th September 3 PM | Tr50+ | 19 |
| 30th April 3 PM | Tr50+ | 7 | 24th June 2 PM | Tr50+ | 31 | 5th October 8 AM | Control+ | 48 |
| 2nd May 9 AM | Control- | 44 | 21st July 9 AM | Control+ | 39 | 5th October 10 AM | Tr1- | 18 |
| 2nd May 11 AM | Tr1+ | 2 | 21st July 12 PM | Control- | 43 | 5th October 12 PM | Control+ | 46 |
| 2nd May 2 PM | Control+ | 40 | 21st July 2 PM | Tr1+ | 1 | 5th October 2 PM | Control- | 52 |
| 2nd May 2 PM | Control+ | 38 | 21st July 5 PM | Tr1- | 5 | 6th October 8 AM | Tr1+ | 14 |
| 3rd May 9 AM | Control+ | 47 | 21st July 9 AM | Control+ | 37 | 6th October 10 AM | Tr50+ | 19 |
| 3rd May 9 AM | Control- | 51 | 22nd July 12 PM | Tr1+ | 2 | 6th October 12 PM | Tr50- | 23 |
| 3rd May 12 PM | Control+ | 45 | 22nd July 2 PM | Tr1- | 6 | 6th October 2 PM | Control+ | 53 |
| 3rd May 12 PM | Tr1+ | 13 | 22nd July 5 PM | Control- | 44 | 6th October 4 PM | Control+ | 55 |
| 8th May 9 AM | Tr1- | 18 | 23rd July 8 AM | Control+ | 38 | 7th October 8 AM | Control+ | 56 |
| 8th May 9 AM | Control+ | 48 | 23rd July 11 AM | Control+ | 40 | 7th October 10 AM | Tr1+ | 25 |
| 8th May 12 PM | Control- | 52 | 23rd July 1 PM | Control- | 51 | 7th October 12 PM | Tr1- | 29 |
| 8th May 12 PM | Tr1+ | 14 | 23rd July 4 PM | Tr1- | 17 | 7th October 2 PM | Control- | 59 |
| 8th May 2 PM | Tr50+ | 19 | 23rd July 6 PM | Tr1+ | 13 | 8th October 8 AM | Control- | 60 |



| | | | | | | | | |
|---|---|---|---|---|---|---|---|---|
| 9th May 12 PM | Control+ | 53 | 24th July 6 AM | Control+ | 47 | 8th October 10 AM | Control+ | 54 |
| 9th May 12 PM | Tr1+ | 26 | 24th July 9 AM | Control+ | 45 | 8th October 12 PM | Tr1- | 30 |
| 10th May 8 AM | Control- | 60 | 24th July 11 AM | Tr50+ | 19 | 8th October 3 PM | Tr1+ | 26 |
| 10th May 11 AM | Control+ | 56 | 17th August 10 AM | Tr1- | 18 | 9th October 8 AM | Tr50- | 35 |
| 10th May 11 AM | Tr1- | 30 | 17th August 12 PM | Control+ | 48 | 9th October 9 AM | Tr50+ | 31 |
| 10th May 1 PM | Tr50- | 35 | 17th August 3 PM | Control- | 52 | 19th October 7 PM | Control- | 43 |
| 10th May 2 PM | Tr50+ | 31 | 18th August 9 AM | Tr1+ | 14 | 19th October 12 PM | Tr1+ | 1 |
| 21st May 10 AM | Control- | 43 | 18th August 11 AM | Control+ | 46 | 19th October 2 PM | Control+ | 39 |
| 21st May 10 AM | Control+ | 37 | 18th August 2 PM | Tr50+ | 19 | 20th October 8 AM | Control+ | 37 |
| 21st May 1PM | Tr1- | 5 | 18th August 5 PM | Tr50- | 23 | 20th October 10 AM | Tr50+ | 7 |
| 21st May 1 PM | Control- | 43 | 19th August 9 AM | Control+ | 55 | 20th October 1 PM | Tr1- | 5 |
| 21st May 4 PM | Control+ | 39 | 19th August 11 AM | Control- | 59 | 20th October 3 PM | Tr50- | 11 |
| 22nd May 9 AM | Control+ | 40 | 19th August 2 PM | Tr1+ | 25 | 21st October 8 AM | Tr1+ | 2 |
| 22nd May 9 AM | Tr1+ | 2 | 19th August 4 PM | Tr1- | 29 | 21st October 10 AM | Tr1- | 6 |
| 22nd May 11 AM | Tr- | 6 | 20th August 9 AM | Tr50+ | 31 | 21st October 12 PM | Tr1- | 5 |
| 22nd May 12 PM | Control- | 44 | 20th August 12 AM | Control+ | 53 | 21st October 1 PM | Control+ | 38 |
| 22nd May 2 PM | Control+ | 38 | 20th August 2 PM | Control+ | 56 | 21st October 3 PM | Control- | 44 |
| 22nd May 2 PM | Tr1+ | 1 | 21st August 9 AM | Control- | 60 | 22nd October 8 AM | Tr1- | 17 |
| 23rd May 10 AM | Tr1+ | 13 | 21st August 11 AM | Tr1+ | 26 | 22nd October 10 AM | Control+ | 45 |
| 23rd May 1 PM | Control+ | 45 | 21st August 4 PM | Control+ | 54 | 22nd October 12 PM | Control- | 51 |
| 23rd May 1 PM | Control- | 51 | 31st August 8 AM | Tr1+ | 1 | 22nd October 3 PM | Tr1+ | 13 |
| 23rd May 3 PM | Tr1- | 17 | 31st August 10 AM | Tr1- | 5 | 23rd October 8 AM | Control+ | 47 |
| | | | | | | 23rd October 10 AM | Tr50+ | 19 |

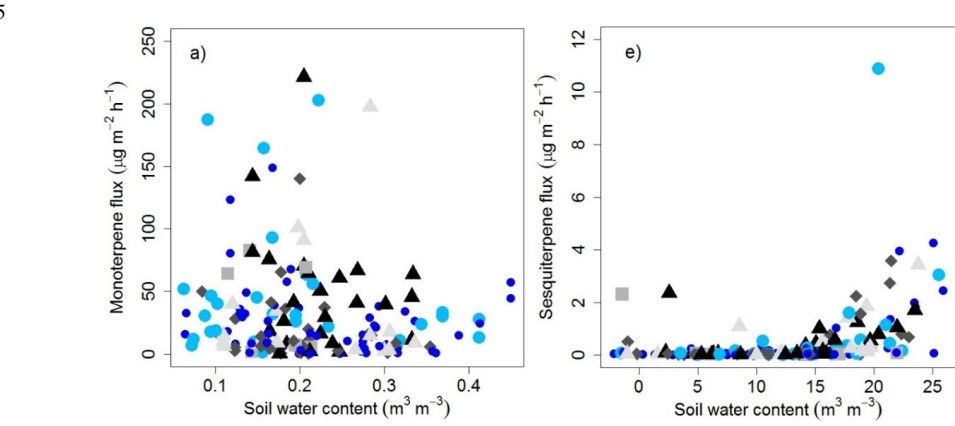



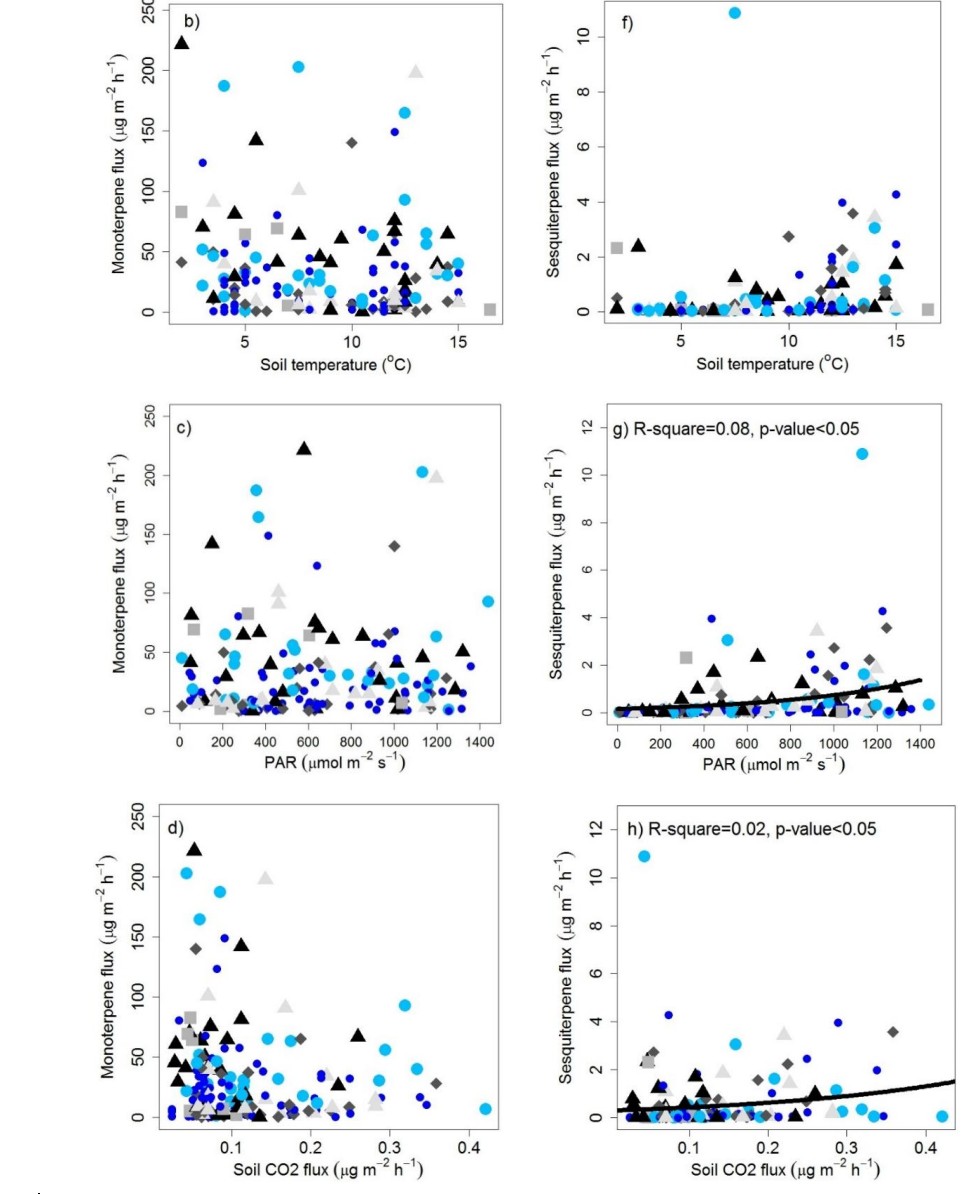

**Figure A1.** Relationships between monoterpene (a–d) and sesquiterpene flux (e–h) ($\mu$g m$^{-2}$ h$^{-1}$) and soil water content (m$^3$ m$^{-3}$), soil temperature (°C), above-canopy PAR ($\mu$mol m$^{-2}$ s$^{-1}$), and soil $CO_2$ flux ($\mu$g m$^{-2}$ h$^{-1}$). The presented data were combined from all treatments (Control: soil was non-trenched, Tr50: the ingrowth of mycorrhizal fungi was allowed, and Tr1: decomposers were the only source) with vegetation (+) and those with bare soil (-). The treatments (Control+=small blue circle, Control-=solid blue circle, Tr50+=filled gray triangle point-up, Tr50-=filled gray square, Tr1+=filled gray diamond,





and Tr1-=filled black triangle point-up) were measured from April to October 2015. The regression coefficient and p-value are indicated where the regression was significant.

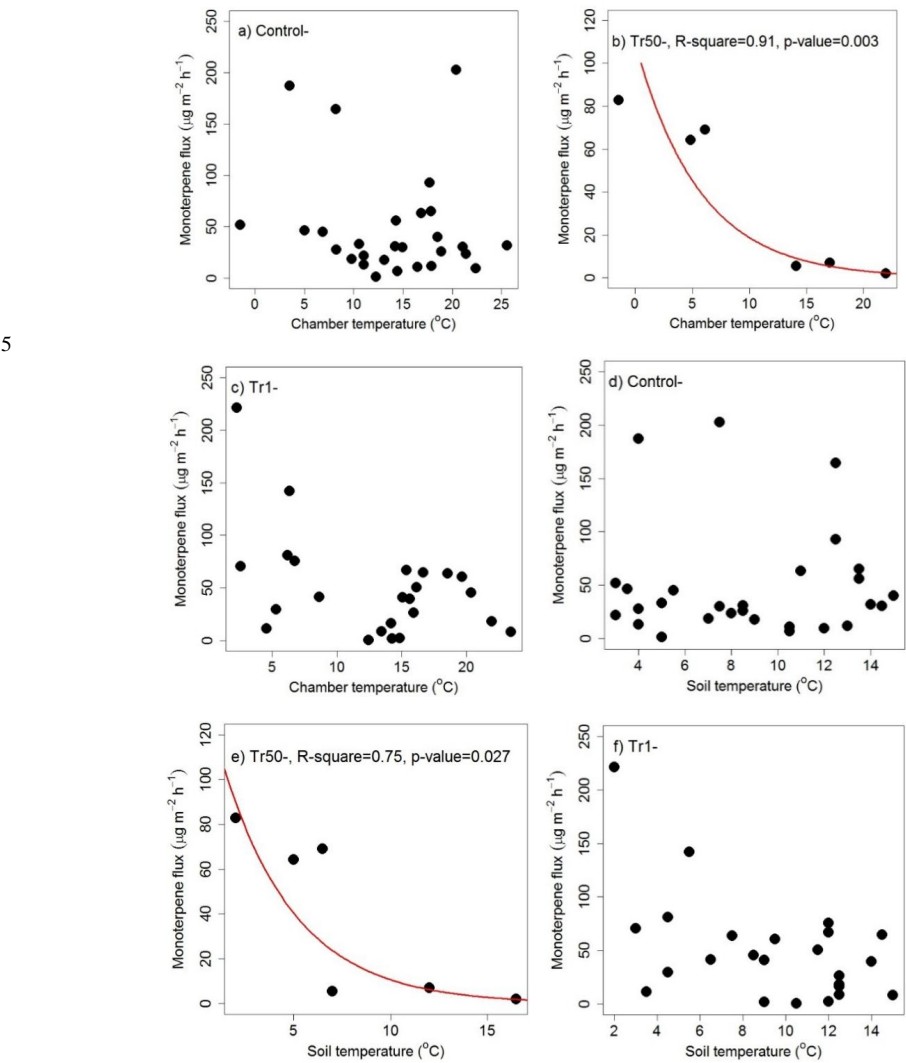

**Figure A2**. Relationships between monoterpene flux ($\mu$g m$^{-2}$ h$^{-1}$) and chamber temperature ($^{o}$C) on Control- (bare soil was non-trenched), Tr50- (bare soil where ingrowth of mycorrhizal fungi was allowed) and Tr1- (bare soil where decomposers were the only source) plots (a, b and c) and soil temperature ($^{o}$C) on Control-, Tr50- and Tr1- plots (d, e and f). The presented data were combined from all the periods, 2015. The regression coefficient and p-value are indicated where the regression was significant.