# Peer review of "Contribution of understorey vegetation and soil processes to boreal forest isoprenoid exchange"

_Biogeosciences, 2016_

## Referee Comment (RC1) · A. B. Guenther (Referee) · 28 Sep 2016

This manuscript describes a field study characterizing VOC emissions from a boreal forest floor. The study provides valuable new observations and insights. A novel aspect of the study is their approach to segregate roots from the rest of the system. The paper is well-written and this is an important topic of general interest to readers of Biogeosciences. I recommend the paper be published after the authors address the following points:

General: The text indicates that these emissions are an important component of forest emissions (for example, Page 2, line 35, Page 14, line 8, Conclusions section) but the authors have not really made the case for this. They do show that these emissions

become relatively more important in spring and fall but they are still small so the importance is not clear. In order to conclude that this is important, and should be the focus of future studies, the authors should provide some quantitative evidence that these low level emissions are significant with respect to their impact on atmospheric composition. This would also enhance the impact of this manuscript. Perhaps this could be done with a simple 1D modeling study or even referencing past studies that have already been conducted at this well studied site.

Specific: Page 2, line 33: While this statement is generally correct, it should be noted there is a wide range of solubility and reactivity for different terpenoid compounds.

Page 3, line 3: Clarify whether you mean that it changes the flux measured with an enclosure or the actual flux

Page 4, line 24: The third point is an objective but not a hypothesis

Page 5, line 1: what is the tree cover fraction at this site?

Section 2.2: Some analytical details should be given including the precision and accuracy of the flux measurements and whether there were any replicate samples to test the reproducibility of the tubes. How was the methanol flushed away? Were tests done to ensure that none of the VOC standards were removed in the process?

Section 2.3: the detection limit should also consider the detection limit of the VOC quantification.

Section 3.2: It is a bit difficult to follow the text in this section. I am not sure what is meant by the second sentence. Also, it is stated that understory vegetation is a monoterpene sink but then goes on to indicate that there was no difference when vegetation was present as long as there were fungi. If the presence of the fungi is the typical situation then this suggests that the vegetation is not a sink.

Section 3.3: The title of this section suggests this will focus on soil sources but instead it discusses vegetation which was the focus of the previous section.

Page 10. Line 4: rewrite the sentence to clarify what was observed in October. Was it high isoprene or high temperature/PAR?

Page 13, line 6 to 11: An alternative hypothesis is that the VOC are consumed by microbes living on the leaves. It seems to me that this just as likely as the possibility that they are absorbed on the cuticle.

Page 14, line 7: define/quantify what you mean by "decent"

Page 14, line 8: What is meant by "unsolved"

Page 14, line 30: How does this overcome the issue of measuring net exchange? The fast response instrument will still be measuring net exchange.

---

## Referee Comment (RC2) · Anonymous Referee #3 · 29 Nov 2016

The manuscript presents an interesting study of the influence of understory on boreal forest isoprenoid fluxes, including an approach enabling investigation of separate factors such as vegetation vs. bare soil and in-growth of microbes and fungi. Generally, the manuscript, including figures and tables, presents the results in a clear and straightforward way. I have listed my specific comments below.

Specific comments:

Abstract. Most sentences start with "we" or "our". Please try to vary this. In line 17 "Our results show that" can be removed.

Page 2 lines 27-32: This is a very long sentence with a lot of information. Please

rephrase.

Page 3 Line 9: "Photosynthesized carbon through the roots was shown to currently contribute 54% of soil respiration". Please clarify what you mean here.

P3 L12: "The main monoterpene sources are suggested to degrade litter" Do you mean "to be degraded litter"?

P3L21-22: rephrase to "from primary metabolism and energy generation of decomposers".

P4 L15-16: Do you mean fluxes from soil?

P6 L24: Remove "&". Please state details of the GC-MS method including at least column and temperature program.

P8 L19: Information about where the measurements were situated should be moved to experimental section. How far away were these measurements from the study area?

P8 L27: I suggest to explain the abbreviations for the study areas the first time they appear in the text.

P8 section 3.1: It would be useful for the reader if you spend some time in the first section to give an overview of the data set such as ranges of fluxes, before discussing correlations. This could include moving some text from 3.2 to 3.1.

P9 L6: discovered -> observed.

P9 L13: "Instead" does not seem like the right word here.

P9 L13-19: Please try to keep sentences about the same group of compounds together, to improve readability.

P11 L5-10: This can be removed since this is clear from the previous sections.

P11 L35: significant -> considerable (unless the authors did a statistical test of this).

P12 L7: Remove "references".

P13 L8: "absorbed" should be changed to "adsorbed".

P14L19: "disappear" is not the right word here. Use "be removed" or similar instead.

P15 L1-2: Could the fan affect the removal rate/deposition of VOC in your chamber?

---

## Author Comment (AC1) · 27 Dec 2016

We would like to express our gratitude for the Reviewers and Editor Marianne Glasius for taking your valuable time to evaluate this manuscript. We thank Alex Guenther for considering this manuscript as generally interesting to readers of Biogeosciences and we thank both Reviewers for the relevant and constructive suggestions which have guided us to make improvements to the manuscript. We have carefully considered the suggestions and revised the manuscript based on the feedback. In the following text we will respond to the reviewer's suggestions (reviewer's suggestions numbered in black font and our response below each comment in green).

Reviewer's suggestions:

1) Reviewer #1: This manuscript describes a field study characterizing VOC emissions from a boreal forest floor. The study provides valuable new observations and insights. A novel aspect of the study is their approach to segregate roots from the rest of the system. The paper is well-written and this is an important topic of general interest to readers of Biogeosciences. I recommend the paper be published after the authors address the following points:

General: The text indicates that these emissions are an important component of forest emissions (for example, Page 2, line 35, Page 14, line 8, Conclusions section) but the authors have not really made the case for this. They do show that these emissions become relatively more important in spring and fall but they are still small so the importance is not clear. In order to conclude that this is important, and should be the focus of future studies, the authors should provide some quantitative evidence that these low level emissions are significant with respect to their impact on atmospheric composition. This would also enhance the impact of this manuscript. Perhaps this could be done with a simple 1D modeling study or even referencing past studies that have already been conducted at this well studied site.

Previous analyses from the same site show that the magnitude of soil and understorey monoterpene emissions in pine forest is rather variable in time, but that in maximum it can make up to 10-15% of the ecosystem scale emissions (Aaltonen et al., 2013). While Aaltonen et al., only report the emissions of monoterpenes, we also emphasize that the forest floor VOC exchange is relevant since the current knowledge of sesquiterpene exchange from the boreal forest floor is very limited.

"Sesquiterpene emissions can be significantly higher than the currently measured flux rates since they are difficult to detect and quantify due to the low volatility and high reactivity (Guenther et al., 2013). Sesquiterpenes are important in the atmospheric processes since they have high precursor potential for secondary organic aerosol (SOA) formation (Guenther et al., 2011)." Page 15, lines 13-17:

The large soil source for reactive compounds may also explain the missing OH sink in the canopy layer (Sinha et al., 2010, Nölscher et al., 2012).

Aaltonen H., Aalto J., Kolari P., Pihlatie M., Pumpanen J., Kulmala M., Nikinmaa E., Vesala T., and Bäck J.: Continuous VOC flux measurements on boreal forest floor. Plant and Soil 369, 241–256, doi:10.1007/s11104-012-1553-4, 2013.

Guenther, A.: Biological and chemical diversity of biogenic volatile organic fluxes into the atmosphere. ISRN Atmospheric Sciences, Volume 2013, doi:10.1155/2013/786290, 2013.

Guenther, A., Kulmala, M., Turnipseed, A., Rinne, J., Suni, T., and Reissell, A.: Integrated land ecosystem-atmosphere processes study (iLEAPS) assessment of global observational networks.

Boreal Environment Research, vol. 16, no. 4, pp. 321–336, 2011.

Nölscher, A. C., Williams, J., Sinha, V., Custer, T., Song, W., Johnson, A. M., Axinte, R., Bozem, H., Fischer, H., Pouvesle, N., Phillips, G., Crowley, J. N., Rantala, P., Rinne, J., Kulmala, M., Gonzales, D., Valverde-Canossa, J., Vogel, A., Hoffmann, T., Ouwersloot, H. G., Vilà-Guerau de Arellano, J., and Lelieveld, J.: Summertime total OH reactivity measurements from boreal forest during HUMPPA-COPEC 2010. Atmospheric chemistry and Physics 12, no. 17: 8257-8270, 2012.

Sinha, V., Williams, J., Lelieveld, J., Ruuskanen, T.M., Kajos, M.K., Patokoski, J., Hellen, H., Hakola, H., Mogensen, D., Boy, M. and Rinne, J.: OH reactivity measurements within a boreal forest: evidence for unknown reactive emissions. Environmental science & technology, 44(17), pp.6614-6620, 2010.

2) Reviewer #1: Specific: Page 2, line 33: While this statement is generally correct, it should be noted there is a wide range of solubility and reactivity for different terpenoid compounds.

We strongly agree that terpenoids are a large group of compounds with different chemical properties, including atmospheric lifetime and solubility, and we have clarified this (Page 3, lines 1-4):

"Isoprenoids are very diverse group of chemical species (Guenther, 2013). Daytime lifetimes of BVOCs in the ambient air varies from minutes (sesquiterpenes) to hours (isoprene, monoterpenes) (Rinne et al., 2007, Bouvier-Brown, 2009; Guenther, 2013, Peräkylä et al., 2014)."

In Section 4.4 we have discussed this issue by introducing lifetime of different isoprenoids. Page 15, lines 17-19: "Sesquiterpene flux rates are probably underestimated more than isoprene and monoterpene flux rates, since daytime lifetime (OH and O3) is 1.3 min for β-caryophyllene, 27 min for isoprene, 29 min for Δ3-carene, and 41 min for α-pinene (Rinne et al., 2007)."

Guenther, A.: Biological and chemical diversity of biogenic volatile organic fluxes into the atmosphere. ISRN Atmospheric Sciences, Volume 2013, doi:10.1155/2013/786290, 2013.

Bouvier-Brown, N. C., Goldstein, A. H., Gilman, J. B., Kuster, W. C., and. de Gouw, J. A.:In-situ ambient quantification of monoterpenes, sesquiterpenes and related oxygenated compounds during BEARPEX 2007: implications for gas- and particle-phase chemistry. Atmospheric Chemistry and Physics, vol. 9, no. 15, pp. 5505–5518, 2009.

Peräkylä, O., Vogt, M., Tikkanen, O. P., Laurila, T., Kajos, M. K., Rantala, P. A., Patokoski, J., Aalto, J., Yli-Juuti, T., Ehn, M., Sipilä, M., Paasonen, P., Rissanen, M., Nieminen, T., Taipale, R., Keronen, P., Lappalainen, H. K., Ruuskanen, T. M., Rinne, J. Kerminen, V.M., Kulmala, M., Bäck, J., Petäjä, T.: Monoterpenes' oxidation capacity and rate over a boreal forest: temporal variation and connection to growth of newly formed particles. *Boreal Environment Research*, *19*, 293-293, 2014.

Rinne, J., Taipale, R., Markkanen, T., Ruuskanen, T. M., Hellén, H., Kajos, M. K., Vesala, T., and Kulmala, M.:Hydrocarbon fluxes above a Scots pine forest canopy: measurements and modeling. Atmospheric Chemistry and Physics, 7(12), 3361–3372, doi:10.5194/acp-7-3361-2007, 2007.

3) Reviewer #1: Page 3, line 3: Clarify whether you mean that it changes the flux measured with an enclosure or the actual flux

The sentence has been clarified on Page 3, lines 8-11: "Large biomass or coverage of understorey vegetation can also decrease the total measured VOC flux from soil because transpiration can induce the formation of water film on the leaf and chamber inner surfaces, which can enhance isoprenoid absorption."

4) Reviewer #1: Page 4, line 24: The third point is an objective but not a hypothesis

We have rewritten this sentence more precisely (Page 4, lines 31-33): "A statistical model including prevailing temperature, seasonality, trenching treatments, understorey vegetation cover, above-canopy PAR, soil water content, and soil temperature can be used to estimate isoprenoid fluxes."

5) Reviewer #1: Page 5, line 1: what is the tree cover fraction at this site?

The stem basal area of all the trees was added on Page 5, lines 8-9.

6) Reviewer #1: Section 2.2: Some analytical details should be given including the precision and accuracy of the flux measurements and whether there were any replicate samples to test the reproducibility of the tubes. How was the methanol flushed away? Were tests done to ensure that none of the VOC standards were removed in the process?

We have presented total uncertainty for the emissions on Page 15, lines 22-24, and calculations and results of total uncertainty, precision and systematic errors for the emissions on Pages 38 and 39 (Appendix, Table A2).

"Methanol was flushed away using nitrogen ($N_2$) flow of 80 ml min$^{-1}$ through the Tenax TA-Carboback-B adsorbent tubes for 10 minutes." This description and more analytical details have now been added on Page 6, lines 28-36. Flow and time for nitrogen flushing have been optimized and no losses have been detected within 10 minutes flushing with the flow of ~80 ml/min for any of the studied compounds. Breakthrough volumes are much higher, in the order of hours with this flow.

7) Reviewer #1: Section 2.3: the detection limit should also consider the detection limit of the VOC quantification.

We fully agree that the term 'the detection limit of the VOC quantification' is better in this context. We have added this term into the Sections 2.2 (Page 7, line 1) and 2.3 (Page 7, line 14), and into Table 3 (Page 30).

8) Reviewer #1: Section 3.2: It is a bit difficult to follow the text in this section. I am not sure what is meant by the second sentence. Also, it is stated that understory vegetation is a monoterpene sink but then goes on to indicate that there was no difference when vegetation was present as long as there were fungi. If the presence of the fungi is the typical situation then this suggests that the vegetation is not a sink.

The second sentence was rewritten on Page 9, line 35.

The sum of the monoterpene fluxes was higher from bare soil than from soil with vegetation cover, where the ingrowth of mycorrhizal fungi was allowed, but the difference was not statistically significant due to the high variation of the emissions. The dominating compound was α-pinene. Different mycorrhizal fungal species produce different amounts of α-pinene (Bäck et al., 2010). One explanation for this would be that the experimental plots included mycorrhizal fungal species which

produce high amounts of α-pinene and this would decrease flux differences between bare soil and soil with vegetation cover, when the ingrowth of mycorrhizal fungi was allowed.

9) Reviewer #1: Section 3.3: The title of this section suggests this will focus on soil sources but instead it discusses vegetation which was the focus of the previous section.

We agree with the Reviewer, and for this reason the sections 3.2 and 3.3 were combined into one section (Pages 9-10).

10) Reviewer #1: Page 10. Line 4: rewrite the sentence to clarify what was observed in October. Was it high isoprene or high temperature/PAR?

We agree that this sentence is unclear and reader has to interpret the message. We have rewritten the sentence and it has been added on Page 10, lines 29-30: "Isoprene fluxes were highest in June and July when temperature and PAR was high (Fig. 3), but interestingly high isoprene fluxes were also observed in October, when temperature and PAR was low."

11) Reviewer #1: Page 13, line 6 to 11: An alternative hypothesis is that the VOC are consumed by microbes living on the leaves. It seems to me that this just as likely as the possibility that they are absorbed on the cuticle.

Several studies have been published which support our conclusion that hydrocarbons can be adsorbed on lipophilic layer on plant leaves (Brown et al., 1998, Welke et al., 1998, Binnie et al., 2002, Joensuu et al., 2016). We think that this is a very interesting suggestion from the reviewer and we have discussed it carefully. This alternative conclusion was added on Page 13, line 35.

Binnie, J., Cape, J. N., Mackie, N., and Leith, I. D.: Exchange of organic solvents between the atmosphere and grass – the use of open top chambers, Sci. Total Environ., 285, 53–67, 2002.

Brown, R. H. A., Cape, J. N., and Farmer, J. G.: Partitioning of chlorinated solvents between pine needles and air, Chemosphere, 36, 1799–1680, 1998.

Farré-Armengol, G., Filella, I., Llusia, J., and Peñuelas, J.: Bidirectional Interaction between Phyllospheric Microbiotas and Plant Volatile Emissions. Trends in Plant Science, 21(10), 854–860, 2016.

Joensuu, J., Altimir, N., Hakola, H. Rostás, M., Raivonen, M., Vestenius, M., Aaltonen, H., Riederer, M., and Bäck J. Role of needle surface waxes in dynamic exchange of mono- and sesquiterpenes. - Atmospheric Chemistry and Physics Discussions doi:10.5194/acp-2015–1024, 2016.

Welke, B., Ettlinger, K., and Riederer, M.: Sorption of volatile organic chemicals in plant surfaces, Environ. Sci. Technol., 32, 1099–1104, 1998.

12) Reviewer #1: Page 14, line 7: define/quantify what you mean by "decent"

This sentence was rewritten more carefully on Page 14, lines 34-35.

13) Reviewer #1: Page 14, line 8: What is meant by "unsolved"

The sentence was rephrased (Page 15, lines 34-36 and Page 15, lines 1-3):

"The mixed effects linear models explained considerable part (43%) of variation in monoterpene emissions although more improvement should be achieved in the future. Possible reasons behind the emissions not explained by the model are oxygen and nutrient availability (Rinnan et al., 2011, the fertilization effect of *Salix phylicifolia* on the β-selinene flux), quality and quantity of the organic matter, soil composition, and microbial community structure, which were not determined in our study."

14) Reviewer #1: Page 14, line 30: How does this overcome the issue of measuring net exchange? The fast response instrument will still be measuring net exchange.

This paragraph was changed so that with PTR-MS it is possible to follow fast changes in the emission, but we chose to use TD-GC-MS which enables the speciation of different compounds (Page 15, lines 27-32).

15) Reviewer 2#: The manuscript presents an interesting study of the influence of understory on boreal forest isoprenoid fluxes, including an approach enabling investigation of separate factors such as vegetation vs. bare soil and in-growth of microbes and fungi. Generally, the manuscript, including figures and tables, presents the results in a clear and straightforward way. I have listed my specific comments below.

16) Reviewer #2: Abstract. Most sentences start with "we" or "our". Please try to vary this. In line 17 "Our results show that" can be removed.

We rephrased the sentences (Page 2, lines 4-21) as suggested.

17) Reviewer #2: Page 2 lines 27-32: This is a very long sentence with a lot of information. Please rephrase.

This very long sentence was rewritten (Page 2 lines 29-33) in the following way:
"The boreal forest floor, including tree roots, understorey vegetation (grasses, shrubs, mosses, lichens, and other vegetation) and the organic soil layer (different stages of decomposing litter, a variety of decomposing and other microorganisms) emits isoprenoids. According to the earlier studies, the boreal forest floor emits monoterpenes (Aaltonen et al., 2011, 5 µg m$^{-2}$ h$^{-1}$ and Hellén et al., 2006, 0–373 µg m$^{-2}$ h$^{-1}$), isoprene (Aaltonen et al., 2011, 0.050 µg m$^{-2}$ h$^{-1}$ and Hellén et al., 2006, 0–1.9) and sesquiterpenes (Aaltonen et al., 2011, 0.045 µg m$^{-2}$ h$^{-1}$ and Hellén et al., 2006 0–0.8 µg m$^{-2}$ h$^{-1}$: β-caryophyllene)."

18) Reviewer #2: Page 3 Line 9: "Photosynthesized carbon through the roots was shown to currently contribute 54% of soil respiration". Please clarify what you mean here.

We agree that this sentence should be written more carefully. The sentence was rephrased on Page 3 (lines 15-16) by writing that "Photosynthesized carbon allocated belowground was shown to contribute 54% of soil respiration".

19) Reviewer #2: P3 L12: "The main monoterpene sources are suggested to degrade litter" Do you mean "to be degraded litter"?

The expression "degrade litter" was rewritten to "degraded litter" (Page 3, line 19).

20) Reviewer #2: P3L21-22: rephrase to "from primary metabolism and energy generation of decomposers".

Corrected as suggested (Page 3, line 28).

21) Reviewer #2: P4 L15-16: Do you mean fluxes from soil?

We agree that the reference to the soil fluxes was not clear enough so the sentence was rephrased according to the reviewer's suggestion by writing on Page 4 (lines 22-23) that "high isoprenoid fluxes from soils are also measured after rain events (Greenberg et al., 2012)".

22) Reviewer #2: P6 L24: Remove "&". Please state details of the GC-MS method including at least column and temperature program.

"&" was removed by describing a mass selective detector in the following way "Perkin Elmer Clarus 600T, Waltham, USA" (Page 6, line 29). More details of the GC-MS method and temperature program was added (Page 6, line 32-33).

23) Reviewer #2: P8 L19: Information about where the measurements were situated should be moved to experimental section. How far away were these measurements from the study area?

All the measurements (flux measurements and environmental data) were executed at the SMEAR II stand. The sentence "Ambient air temperature and PAR were measured at the SMEAR II stand" on Page 8 (line 24) was situated to the Trenching experiment –section on Page 5 (lines 33-37) in the following way.

"All plots at the SMEAR II stand were equipped with a 0.5m long tube, where soil water content was measured using the capacity probe (PR2, Delta-T Devices) every second week. Soil temperature sensors were placed in the soil surface layer on each plot (depth 4 cm), and data were logged every fourth hour from May to October in 2012–2015. Photosynthetically active radiation (PAR) was measured also measured at the SMEAR II stand from a wavelength range of 400–700 nm using an LI-19OSZ quantum sensor (Li-Cor, Biosciences, Lincoln, NE) at heights of 18.0 m (above canopy) and 0.6 m (below canopy)."

24) Reviewer #2: P8 L27: I suggest to explain the abbreviations for the study areas the first time they appear in the text.

The reviewer makes an excellent point that the abbreviations should be explained in this section. The sentence was rewritten (Page 9, lines 14-17):
"Chamber and soil temperature did not differ between treatments, except during July and August (period 4), when soil temperature was higher in Control- (13.5 $^{\circ}$C), where the ingrowth of roots and mycorrhizal fungi was allowed without understorey vegetation cover, than in Control+ (12.6 $^{\circ}$C) with understorey vegetation cover. Soil water content was higher in Control+ (0.13 m3 m-3) than in Control- (0.10 $m^3$ $m^{-3}$) and higher in Tr1- (0.19 $m^3$ $m^{-3}$, only decomposer activity was allowed without understorey vegetation cover) compared to Control+ and Control- in September and early October (period 5) (data not shown)."

25) Reviewer #2: P8 section 3.1: It would be useful for the reader if you spend some time in the first section to give an overview of the data set such as ranges of fluxes, before discussing correlations. This could include moving some text from 3.2 to 3.1.

We wrote a new section where we give an overview about which compounds were measured and which were the dominating compounds. We will also give flux range of isoprene, monoterpenes and sesquiterpenes for the whole data (Page 8, lines 29-35 and Page 9, lines 1-3).

26) Reviewer #2: P9 L6: discovered -> observed.

The more accurate expression "observed" was used (Page 9, line 29).

27) Reviewer #2: P9 L13: "Instead" does not seem like the right word here.

We have rephrased the sentence according to reviewer's suggestions (Page 10, line 2).

28) Reviewer #2: P9 L13-19: Please try to keep sentences about the same group of compounds together, to improve readability.

We very much agree with the reviewer and the order of the sentences were reorganized in the following way (Page 8, lines 31-35 and Page 9, lines 1-3) : "Monoterpene flux range was 0.40–221.0 $\mu$g m$^{-2}$ h$^{-1}$ (data not shown). The most dominating compounds were $\alpha$-pinene, camphene, $\beta$-pinene, and $\Delta^3$-carene, covering 84–94% of the flux spectra (Table 3). The exception was Tr1+, where isoprene covered 20% of the spectrum (Table 3). Sesquiterpene flux range was 0.01–10.9 $\mu$g m$^{-2}$ h$^{-1}$ (data not shown). Sesquiterpene fluxes from various sources were equally low (0.35–0.73 $\mu$g m$^{-2}$ h$^{-1}$), and the most abundant sesquiterpenes emitted were $\beta$-caryophyllene and aromadendrene (Table 3). Isoprene fluxes from the different sources were also low (0.98–4.91 $\mu$g m$^{-2}$ h$^{-1}$) (Table 3) and flux range was 0.005–99.8 $\mu$g m$^{-2}$ h$^{-1}$ (data not shown)."

29) Reviewer #2: P11 L5-10: This can be removed since this is clear from the previous sections.

We believe that it is reader-friendly to shortly summarize the background and aims of the study before going deeper into the interpretation of the results. However, if needed we can also delete this introductory chapter from the discussion.

30) Reviewer #2: P11 L35: significant -> considerable (unless the authors did a statistical test of this).

Litterfall contribution to decomposition processes was not analyzed using a statistical test and for this reason word "considerable" was used (Page 12, line 27) as the reviewer wisely suggested.

31) Reviewer #2: P12 L7: Remove "references".

The word "references" was removed (Page 12, line 34).

32) Reviewer #2: P13 L8: "absorbed" should be changed to "adsorbed".

The misspelling was corrected (Page 13, line 32).

33) Reviewer #2: P14L19: "disappear" is not the right word here. Use "be removed" or similar instead.

The sentence was rephrased by writing "which means that they can be removed through chemical reactions" (Page 15 line 11).

34) Reviewer #2: P15 L1-2: Could the fan affect the removal rate/deposition of VOC in your chamber?

No significant losses were detected for studied compounds in the recovery tests, where fan was used.

[revised manuscript text omitted]

---

## Author Response (AR1)

Dear Editor,

Thank you for your feedback. The following sentence was added on Page 2, lines 33-35, based on the Reviewer's suggestion.

"Soil and understorey monoterpene emissions in pine forest are rather variable in time, but that in maximum they can make up to 10-15% of the ecosystem scale emissions (Aaltonen et al., 2013)."

Sincerely Yours,

Mari Mäki